# Versatile Platinum(IV) Prodrugs of Naproxen and Acemetacin as Chemo-Anti-Inflammatory Agents

**DOI:** 10.3390/cancers15092460

**Published:** 2023-04-26

**Authors:** Angelico D. Aputen, Maria George Elias, Jayne Gilbert, Jennette A. Sakoff, Christopher P. Gordon, Kieran F. Scott, Janice R. Aldrich-Wright

**Affiliations:** 1School of Science, Western Sydney University, Locked Bag 1797, Penrith South, Sydney, NSW 2751, Australia; a.aputen@westernsydney.edu.au (A.D.A.); m.elias3@westernsydney.edu.au (M.G.E.);; 2Ingham Institute, Liverpool, Sydney, NSW 2170, Australia; 3Calvary Mater Newcastle Hospital, Waratah, Newcastle, NSW 2298, Australia; 4School of Medicine, Western Sydney University, Locked Bag 1797, Penrith South, Sydney, NSW 2751, Australia

**Keywords:** platinum(II), cisplatin, chemotherapy, platinum(IV), naproxen, acemetacin, cytotoxicity, ROS, mitochondria, cyclooxygenase-2

## Abstract

**Simple Summary:**

Traditional intravenous platinum(II) chemotherapy drugs such as cisplatin, oxaliplatin and carboplatin are highly effective in treating multiple cancer types. Unfortunately, this treatment is most often beset with detrimental side effects that inevitably impact the patient’s willingness to comply with treatment programs. Platinum(II) drugs are scarcely selective, have poor bioavailability, and exhibit inherent and acquired resistance. A promising approach to address these impediments is the development of kinetically stable octahedral platinum(IV) complexes. This design strategy has been exploited for cisplatin and its derivatives and has been reported widely in the literature. This is a highly attractive approach for synthetic chemists due to the versatility it offers. Here, we contribute to this paradigm shift by using structurally distinct platinum(IV) scaffolds as effective prodrugs. The findings reported are expected to advance our understanding of cancer treatment.

**Abstract:**

Developing new and versatile platinum(IV) complexes that incorporate bioactive moieties is a rapidly evolving research strategy for cancer drug discovery. In this study, six platinum(IV) complexes (**1**–**6**) that are mono-substituted in the axial position with a non-steroidal anti-inflammatory molecule, naproxen or acemetacin, were synthesised. A combination of spectroscopic and spectrometric techniques confirmed the composition and homogeneity of **1**–**6**. The antitumour potential of the resultant complexes was assessed on multiple cell lines and proved to be significantly improved compared with cisplatin, oxaliplatin and carboplatin. The platinum(IV) derivatives conjugated with acemetacin (**5** and **6**) were determined to be the most biologically potent, demonstrating GI_50_ values ranging between 0.22 and 250 nM. Remarkably, in the Du145 prostate cell line, **6** elicited a GI_50_ value of 0.22 nM, which is 5450-fold more potent than cisplatin. A progressive decrease in reactive oxygen species and mitochondrial activity was observed for **1**–**6** in the HT29 colon cell line, up to 72 h. The inhibition of the cyclooxygenase-2 enzyme was also demonstrated by the complexes, confirming that these platinum(IV) complexes may reduce COX-2-dependent inflammation and cancer cell resistance to chemotherapy.

## 1. Introduction

With the rapid expansion of knowledge in coordination and organometallic chemistry, plenty of innovation towards exploiting metal elements to design and create valuable diagnostic and therapeutic drugs have been conveyed in the literature, especially for cancer treatment [1]. Cancer continues to be a major public health concern, and cancer treatment strategies have become more advanced in an attempt to provide effective treatment [2]. Traditional intravenous chemotherapy is still the established clinical regimen and is considered to be the most effective approach to kill cancer cells. This is predominantly due to the ground-breaking clinical success of platinum(II) drugs, *cis*-diamminedichloroplatinum(II) (cisplatin), *trans*-*L-*(1*R*,2*R*-diaminocyclohexane) oxalatoplatinum(II) (oxaliplatin) and *cis*-diammine(1,1-cyclobutanedicarboxylato) platinum(II) (carboplatin) (Figure 1) as mainstream anticancer agents [3,4,5,6]. The therapeutic potential of these drugs is attributed to their ability to covalently bind with deoxyribonucleic acid (DNA) and form crosslinks (i.e., DNA inter/intra-strand crosslinks and DNA-protein crosslinks) that prevent further DNA replication, and consequently, induce apoptosis [5,7,8,9,10]. Although these drugs are effective against multiple tumour types, indiscriminate toxicity, poor bioavailability, and easily acquired drug resistance are challenging parameters that limit their use in the clinic [11,12,13,14].

We have previously reported a class of non-DNA coordinating platinum(II) complexes in the form of **[Pt^II^(H_L_)(A_L_)]^2+^**, where H_L_ is the heterocyclic ligand (i.e., 1,10-phenanthroline (phen), 5-methyl-1,10-phenanthroline (5-Mephen) or 5,6-dimethyl-1,10-phenanthroline (5,6-Me_2_phen)) and A_L_ is an ancillary ligand which is chiral, 1*S*,2*S*-diaminocyclohexane (*SS*-DACH or DACH) (Figure 1), which exhibits exceptional in vitro activity [15,16,17,18,19,20]. This unconventional class of platinum(II) complexes demonstrated potency significantly better than cisplatin, oxaliplatin or carboplatin in multiple cancer cell lines [20,21,22]. Our most potent complex, [Pt^II^(5,6-Me_2_phen)(*SS*-DACH)]^2+^ (**56ME*SS***), demonstrates a mode of action that differs from traditional platinum(II) drugs, as this complex is proposed to modify the cytoskeletal architecture, induce bioenergetic stress that reduces the mitochondrial membrane potential (MtMP) and promotes epigenetic changes [21,23]. This class of platinum(II) complexes exhibits incomparable in vitro activity, but this was not initially seen during in vivo experiments. Treatment with **56ME*SS*** of BD-IX rats exhibiting peritoneal carcinomatosis through intravenous and intraperitoneal routes did not demonstrate tumour suppression response but instead induced nephrotoxicity [24]. Alternatively, when **PHEN*SS*** or cisplatin were administered to mice transplanted with a PC3 (prostate carcinoma) tumour, both complexes exhibited a comparable reduction in mean tumour mass relative to the vehicle-treated control group [19]. Notably, treatment with **PHEN*SS*** did not elicit any toxic side effects, while those mice treated with cisplatin were euthanised due to adverse side effects by Day 20. Due to the in vitro and in vivo discrepancies observed, their corresponding platinum(IV) derivatives have been studied to improve their pharmacokinetics and to potentially enhance in vivo antitumour effects, as demonstrated in preliminary studies [22,25].

Designing platinum(IV) complexes is an effective strategy to improve the pharmacokinetic and pharmacological limitations of platinum(II) complexes [26,27,28,29,30,31,32,33]. The low spin d^6^ electronic configuration of platinum(IV) complexes translates to improved drug stability, which promotes enhanced selectivity with the potential to lessen the incidence of undesirable side effects [26,32]. The six-coordinate octahedral geometry of platinum(IV) complexes allow for the coordination of specialised bioactive or non-bioactive moieties. Platinum(IV) complexes are regularly synthesised via oxidation of platinum(II) complexes in the presence of hydrogen peroxide (H_2_O_2_) or chlorine gas (Cl_2_) to create the core platinum(II) scaffold with two axially coordinated hydroxido (OH) or chlorido (Cl) ligands [31,32], which are available for nucleophilic substitution reactions. 

Intracellularly, platinum(IV) complexes are reported to maintain their structural integrity prior to interacting with their desired biological targets, the cell [34]. Typically, platinum(IV) complexes undergo a two-electron reduction when exposed to biological reductants such as glutathione (GSH) or ascorbic acid (AsA), resulting in the formation of the core platinum(II) species and release of the axial moieties (Figure 2) [26,32,34]. This advantageous key feature allows for the introduction of multiple drugs in a single dose, rather than having to administer combinations of multiple drugs, as exemplified in combination therapy. While drug combinations can maximise the chances of killing cancer cells and overcome resistance to any single drug, it can also complicate the management and treatment of cancer as well as the prediction of therapeutic outcomes because each co-administered drug has a different pharmacokinetic profile and target [35].

The configurational arrangement of platinum(IV) complexes is an attribute of the design and a game changer for researchers in this area of research. Cisplatin, oxaliplatin and carboplatin are the most popular platinum(II) cores used to generate platinum(IV) scaffolds in many studies, and each have been coordinated to various enzymatic inhibitors that supress cancer development and progression [28,36]. In particular, the coordination of non-steroidal anti-inflammatory drugs (NSAIDs) to the cores of cisplatin and its derivatives have resulted in chemo-anti-inflammatory prodrugs that are capable of reversing chemoresistance because of enhanced bioavailability [37,38,39,40,41]. Since the 1970s, NSAIDs have been extensively administered to cancer patients, simply for cancer pain management, but there was also a growing curiosity as to whether the regular ingestion of NSAIDs would help combat cancer and decrease cancer risk [42,43,44,45]. NSAIDs suppress the activity of cyclooxygenase (COX) enzymes, COX-1 and COX-2, which play a role in prostaglandin biosynthesis [46]. Prostaglandins are a group of physiologically active lipids that are involved in cellular signal transduction pathways and also, are key mediators in the inflammatory response [47]. Chronic inflammation is a key factor in cancer progression and occurs in 20% of all human cancers [48]. Because chronic inflammation is directly involved with the overexpression of COX enzymes [49,50], this makes COX enzymes as viable therapeutic targets, especially COX-2, as it is predominantly overexpressed in a few cancer types (i.e., lung, colon, prostate, and breast) [51]. 

We present here six mono-substituted platinum(IV) complexes that incorporate either naproxen (NPX) or acemetacin (ACE), which is the prodrug of indomethacin (Figure 3). The general structure of the synthesised platinum(IV) complexes are defined as **[Pt^IV^(H_L_)(A_L_)(X)(OH)]^2+^**, where X represents NPX and ACE: [Pt^IV^(phen)(*SS*-DACH)(NPX)(OH)]^2+^ (**PHEN*SS*(IV)-NPX** or **1**), [Pt^IV^(5-Mephen)(*SS*-DACH)(NPX)(OH)]^2+^ (**5ME*SS*(IV)-NPX** or **2**), [Pt^IV^(5,6-Me_2_phen)(*SS*-DACH)(NPX)(OH)]^2+^ (**56ME*SS*(IV)-NPX** or **3**), [Pt^IV^(phen)(*SS*-DACH)(ACE)(OH)]^2+^ (**PHEN*SS*(IV)-ACE** or **4**), [Pt^IV^(5-Mephen)(*SS*-DACH)(ACE)(OH)]^2+^ (**5ME*SS*(IV)-ACE** or **5**), and [Pt^IV^(5,6-Me_2_phen)(*SS*-DACH)(ACE)(OH)]^2+^ (**56ME*SS*(IV)-ACE** or **6**) (Figure 3). The axial coordination of NPX or ACE into the platinum(IV) scaffolds is expected to produce prodrugs with enhanced biological activity. This approach will also provide a better understanding of the influence of NSAIDs in platinum(IV) prodrug chemotherapeutics in cancer treatment. 

The composition and homogeneity of the studied platinum(IV) complexes (**1**–**6**) was confirmed by high-performance liquid chromatography (HPLC), nuclear magnetic resonance (^1^H-NMR; two-dimensional correlation spectroscopy (2D-COSY); heteronuclear multiple quantum correlation (^1^H-^195^Pt-HMQC)), ultraviolet–visible (UV), circular dichroism (CD), high-resolution electrospray ionisation mass spectrometry (ESI-MS), and infrared (IR) spectroscopy. The solubility of **1**–**6** was determined. The reduction behaviour of **1**–**6** in aqueous solution containing AsA at 37 °C was reported with the aid of ^1^H-NMR and one-dimensional ^195^Pt-NMR (1D-^195^Pt-NMR). The stability of **1**–**6** in aqueous solution without a reducing agent was monitored using HPLC, at room temperature and at 37 °C for 36 h. Lipophilicity measurements were also undertaken for **1**–**6**, utilising HPLC. In vitro cytotoxicity of **1**–**6** was assessed in multiple cell lines, including: HT29 colon, U87 glioblastoma, MCF7 breast, A2780 ovarian, H460 lung, A431 skin, Du145 prostate, BE2C neuroblastoma, SJG2 glioblastoma, MIA pancreas, ADDP ovarian variant (cisplatin-resistant A2780 clone), and the non-tumour derived MCF10A breast line. Finally, the reactive oxygen species (ROS) potential, changes in MtMP, and COX-2 inhibition capability of **1**–**6** were assessed to evaluate how impactful these parameters are to the overall in vitro cytotoxicity of the complexes. 

## 2. Materials and Methods

### 2.1. Materials

All laboratory reagents used in the experiments were of analytical grade. The deionised water (d.i.H_2_O) utilised for the experiments was procured from a MilliQ^TM^ system (Millipore Australia Pty Ltd., Sydney, NSW, Australia). Phen, 5-Mephen, 5,6-Me_2_phen, NPX, ACE, *N*,*N′*-dicyclohexylcarbodiimide (DCC), dimethyl sulfoxide (DMSO), acetonitrile (CH_3_CN) and trifluoroacetic acid (TFA) were purchased from Sigma-Aldrich, Sydney, NSW, Australia. Sep-Pak^®^ C_18_-reverse phase columns were purchased from Waters Australia Pty Ltd., Sydney, NSW, Australia. Deuterated oxide 99.9% (D_2_O) and deuterated DMSO (DMSO-d_6_) were purchased from Cambridge Isotope Laboratories, Andover, MA, USA. Methanol (MeOH) was obtained from Honeywell Research Chemicals, NJ, USA. Diethyl ether (Et_2_O) and acetone (C_3_H_6_O) were purchased from ChemSupply Australia, Gillman, SA, Australia. Additional laboratory reagents were acquired from commercial sources.

### 2.2. Chemistry

#### 2.2.1. Synthesis Route of NHS Esters

The NHS esters of NPX and ACE were prepared using previously established protocols without modifications (see Appendix A) [52,53,54]. 

**NHS-NPX**—Yield: 250 mg; 82%.^1^H-NMR (400 MHz, DMSO-d_6,_ δ): 7.82 (t, c and d; a, 3H), 7.46 (d, e, 1H, *J* = 8.5 Hz), 7.32 (s, b, 1H), 7.19 (dd, f, 1H, *J*_1_ = 2.1 Hz, *J*_2_ = 8.9 Hz), 4.40 (q, α, 1H, *J* = 7.0 Hz), 3.88 (s, γ, 3H), 2.79 (s, g and h, 4H), and 1.60 (d, β, 3H, *J* = 7.1 Hz). HPLC, T_R_: 254 nm, 11.8 min. ESI-MS: *calculated* for [M]^+^: *m*/*z* = 327.11; *experimental*: *m*/*z* = 327.12.

**NHS-ACE**—Yield: 310 mg; 78%. ^1^H-NMR (400 MHz, DMSO-d_6,_ δ): 7.66 (q, a, b, c and d, 4H, *J* = 8.6 Hz), 7.05 (d, b1, 1H, *J* = 2.4 Hz), 6.94 (d, a1, 1H, *J* = 9.0 Hz), 6.72 (dd, c1, 1H, *J*_1_ = 2.3 Hz, *J*_2_ = 9.0 Hz), 5.20 (s, α1, 2H), 3.94 (s, α2, 2H), 3.76 (s, f, 3H), 2.82 (s, g and h, 4H), and 2.23 (s, e, 3H). HPLC, T_R_: 254 nm, 13.3 min. ESI-MS: *calculated* for [M]^+^: *m*/*z* = 512.10; *experimental*: *m*/*z* = 512.30.

#### 2.2.2. Synthesis Route of Platinum(II) Precursors and Platinum(IV) Scaffolds

All platinum(II) precursors of type **[Pt^II^(H_L_)(A_L_)]^2+^** and platinum(IV) scaffolds of type **[Pt^IV^(H_L_)(A_L_)(OH)_2_]^2+^** were synthesised as reported (see Appendix A) [53,55].

#### 2.2.3. Synthesis Route of Platinum(IV) Derivatives Incorporating NPX and ACE (**1**–**6**)

The synthesis of **[Pt^IV^(H_L_)(A_L_)(X)(OH)]^2+^** was undertaken using previously established protocols (see Appendix A) [52,53].

**1**—Yield: 25 mg; 60%. ^1^H-NMR (400 MHz, D_2_O_,_ δ): 9.12 (t, H2 and H9, 2H, *J =* 5.0 Hz), 8.74 (d, H4, 1H, *J* = 8.3 Hz), 8.70 (d, H7, 1H, *J* = 8.3 Hz), 8.10 (m, H3 and H8, 2H), 7.67 (q, H5 and H6, 2H, *J* = 8.9 Hz), 7.12 (m, c and d, 2H), 7.05 (s, a, 1H), 6.84 (d, e, 1H, *J* = 8.5 Hz), 6.71 (s, b, 1H), 6.40 (dd, f, 1H, *J*_1_ = 1.4 Hz, *J*_2_ = 8.5 Hz), 3.99 (s, γ, 3H), 3.50 (q, α, 1H, *J* = 7.0 Hz), 3.12 (m, H1′ and H2′, 2H), 2.37 (m, H3′ and H6′ eq., 2H), 1.65 (m, H4′ and H5′ eq.; H3′ and H6′ ax., 4H), 1.25 (m, H4′ and H5′ ax., 2H), and 1.15 (d, β, 3H, *J* = 7.1 Hz). ^1^H-^195^Pt-HMQC (400 MHz, D_2_O_,_ δ): 9.12/545 ppm; 8.10/545 ppm. HPLC, T_R_: 254 nm, 7.78 min. UV λ_max_ nm (ε/M.cm^−1^ ± SD × 10^4^, d.i.H_2_O): 206 (8.92 ± 5.54), 227 (7.08 ± 3.59), 279 (2.56 ± 0.25), 306 (0.77 ± 0.88). CD λ_max_ nm (Δε/M.cm^−1^ × 10^1^, d.i.H_2_O): 202 (−1263), 224 (+178), 239 (+329), 281 (−137). IR (cm^−1^): 3380, 3062, 2936, 1603, 1220. ESI-MS: *calculated* for [M-H]^+^: *m*/*z* = 734.23; *experimental*: *m*/*z* = 734.23.

**2**—Yield: 37 mg; 69%. ^1^H-NMR (400 MHz, D_2_O_,_ δ): 9.12 (t, H2, 1H, *J =* 5.6 Hz), 9.04 (d, H9, 1H, *J* = 5.6 Hz), 8.84 (q, H4, 1H, *J* = 8.5 Hz), 8.60 (q, H7, 1H, *J* = 8.3 Hz), 8.10 (m, H3 and H8, 2H), 7.40 (d, H6, 1H, *J =* 19 Hz), 7.10 (d, c and d, 2H, *J =* 3.2 Hz), 7.03 (d, e, 1H, *J* = 4.5 Hz), 6.84 (dd, f, 1H, *J*_1_ = 2.4 Hz, *J*_2_ = 8.5 Hz), 6.67 (d, a, 1H, *J =* 7.8 Hz), 6.40 (d, b, 1H, *J =* 8.4 Hz), 3.97 (s, CH_3_, 3H), 3.50 (m, α, 1H), 3.14 (d, H1′ and H2′, 2H), 2.56 (s, γ, 3H), 2.38 (d, H3′ and H6′ eq., 2H), 1.66 (m, H4′ and H5′ eq.; H3′ and H6′ ax., 4H), 1.27 (m, H4′ and H5′ ax., 2H), and 1.15 (q, β, 3H, *J* = 4.6 Hz). ^1^H-^195^Pt-HMQC (400 MHz, D_2_O_,_ δ): 9.12/544 ppm; 9.04/544 ppm. HPLC, T_R_: 254 nm, 7.98 min. UV λ_max_ nm (ε/M.cm^−1^ ± SD × 10^4^, d.i.H_2_O): 207 (9.95 ± 6.78), 228 (8.39 ± 4.98), 283 (2.92 ± 0.12), 312 (0.86 ± 0.85). CD λ_max_ nm (Δε/M.cm^−1^ × 10^1^, d.i.H_2_O): 201 (−1061), 223 (+120), 232 (+48.1), 243 (+309), 287 (−109). IR (cm^−1^): 3471, 3076, 2940, 1605, 1220. ESI-MS: *calculated* for [M-H]^+^: *m*/*z* = 748.25; *experimental*: *m*/*z* = 748.25.

**3**—Yield: 29 mg; 65%. ^1^H-NMR (400 MHz, D_2_O_,_ δ): 9.06 (d, H2 and H9, 2H, *J =* 5.5 Hz), 8.84 (t, H4 and H7, 2H, *J* = 9.0 Hz), 8.09 (m, H3 and H8, 2H), 7.01 (s, c and d, 2H), 6.94 (s, a, 1H), 6.77 (d, e, 1H, *J* = 8.5 Hz), 6.62 (s, b, 1H), 6.38 (d, f, 1H, *J* = 8.4 Hz), 3.92 (s, γ, 3H), 3.48 (q, α, 1H, *J =* 7.0 Hz), 3.16 (m, H1′ and H2′, 2H), 2.42 (s, 2 × CH_3_; H3′ and H6′ eq., 8H), 1.67 (m, H4′ and H5′ eq.; H3′ and H6′ ax., 4H), 1.28 (m, H4′ and H5′ ax., 2H), and 1.15 (d, β, 3H, *J* = 7.0 Hz). ^1^H-^195^Pt-HMQC (400 MHz, D_2_O_,_ δ): 9.06/534 ppm; 8.09/534 ppm. HPLC, T_R_: 254 nm, 8.19 min. UV λ_max_ nm (ε/M.cm^−1^ ± SD × 10^4^, d.i.H_2_O): 208 (8.73 ± 5.35), 231 (6.74 ± 3.23), 289 (2.08 ± 0.42), 318 (0.67 ± 0.91). CD λ_max_ nm (Δε/M.cm^−1^ × 10^1^, d.i.H_2_O): 204 (−1007), 223 (+9.32), 230 (−9.99), 244 (+316), 293 (−93.2). IR (cm^−1^): 3466, 3076, 2937, 1603, 1218. ESI-MS: *calculated* for [M-H]^+^: *m*/*z* = 762.26; *experimental*: *m*/*z* = 762.26.

**4**—Yield: 31 mg; 62%. ^1^H-NMR (400 MHz, D_2_O_,_ δ): 9.16 (d, H2, 1H, *J* = 5.4 Hz), 9.14 (d, H9, 1H, *J* = 5.5 Hz), 8.68 (q, H4 and H7, 2H, *J =* 5.6 Hz), 8.03 (q, H3 and H8, 2H, *J =* 5.8 Hz), 7.91 (s, H5 and H6, 2H), 7.21 (s, a and b*;* c and d, 4H), 6.69 (d, b1, 1H, *J =* 8.4 Hz), 6.53 (s, a1, 1H), 6.42 (d, c1, 1H, *J =* 8.4 Hz), 4.27 (s, α2, 2H), 3.55 (s, f, 3H), 3.37 (s, α1, 2H), 3.15 (m, H1′ and H2′, 2H), 2.33 (m, H3′ and H6′ eq., 2H), 1.76 (s, e, 3H), 1.65 (m, H4′ and H5′ eq.; H3′ and H6′ ax., 4H), and 1.21 (m, H4′ and H5′ ax., 2H). ^1^H-^195^Pt-HMQC (400 MHz, D_2_O_,_ δ): 9.16/547 ppm; 9.14/547 ppm; 8.03/547 ppm. HPLC, T_R_: 254 nm, 9.62 min. UV λ_max_ nm (ε/M.cm^−1^ ± SD × 10^4^, d.i.H_2_O): 203 (4.43 ± 0.42), 278 (1.15 ± 2.70), 305 (0.49 ± 3.40). CD λ_max_ nm (Δε/M.cm^−1^ × 10^1^, d.i.H_2_O): 206 (−447), 249 (+8.15). IR (cm^−1^): 3410, 3063, 2942, 1609, 1262, 845. ESI-MS: *calculated* for [M-H]^+^: *m*/*z* = 919.22; *experimental*: *m*/*z* = 920.22.

**5**—Yield: 42 mg; 68%. ^1^H-NMR (400 MHz, D_2_O_,_ δ): 9.18 (dd, H2, 1H, *J* = 5.5 Hz), 9.06 (dd, H9, 1H, *J* = 5.4 Hz), 8.79 (q, H4, 1H, *J =* 5.0 Hz), 8.41 (q, H7, 1H, *J =* 8.5 Hz), 8.08 (t, H3, 1H, *J =* 6.8 Hz), 7.92 (m, H8, 1H), 7.58 (d, H6, 1H, *J =* 6.4 Hz), 7.05 (q, a and b*;* c and d, 4H, *J =* 8.0 Hz), 6.58 (d, b1, 1H, *J =* 8.7 Hz), 6.52 (s, a1, 1H), 6.27 (d, c1, 1H, *J =* 8.6 Hz), 4.28 (m, α2, 2H), 3.48 (s, f, 3H), 3.34 (s, α1, 2H), 3.15 (d, H1′ and H2′, 2H), 2.50 (s, CH_3_, 3H), 2.34 (d, H3′ and H6′ eq., 2H), 1.65 (m, H4′ and H5′ eq.; H3′ and H6′ ax.; e, 7H), and 1.23 (m, H4′ and H5′ ax., 2H). ^1^H-^195^Pt-HMQC (400 MHz, D_2_O_,_ δ): 9.18/545 ppm; 9.06/547 ppm; 8.08/545 ppm; 7.92/545 ppm. HPLC, T_R_: 254 nm, 9.83 min. UV λ_max_ nm (ε/M.cm^−1^ ± SD × 10^4^, d.i.H_2_O): 204 (18.6 ± 5.44), 284 (6.16 ± 2.63), 311 (2.12 ± 1.68). CD λ_max_ nm (Δε/M.cm^−1^ × 10^1^, d.i.H_2_O): 205 (−373), 209 (−419), 234 (−144). IR (cm^−1^): 3385, 3066, 2943, 1608, 1265, 846. ESI-MS: *calculated* for [M-H]^+^: *m*/*z* = 933.23; *experimental*: *m*/*z* = 934.22.

**6**—Yield: 28 mg; 58%. ^1^H-NMR (400 MHz, D_2_O_,_ δ): 9.12 (d, H2, 1H, *J* = 5 Hz), 9.06 (d, H9, 1H, *J* = 5.4 Hz), 8.76 (q, H4 and H7, 2H, *J =* 4.9 Hz), 7.98 (m, H3 and H8, 1H), 6.95 (s, a and b*;* c and d, 4H), 6.53 (s, a1; b1, 2H), 6.18 (d, c1, 1H, *J =* 7.6 Hz), 4.29 (s, α2, 2H), 3.44 (s, *f*, 3H), 3.33 (s, α1, 2H), 3.14 (m, H1′ and H2′, 2H), 2.38 (s, H3′ and H6′ eq.; 2 × CH_3_, 8H), 1.62 (m, H4′ and H5′ eq.; H3′ and H6′ ax.; e, 7H), and 1.23 (m, H4′ and H5′ ax., 2H). ^1^H-^195^Pt-HMQC (400 MHz, D_2_O_,_ δ): 9.12/531 ppm; 9.06/531 ppm; 7.98/531 ppm. HPLC, T_R_: 254 nm, 10.2 min. UV λ_max_ nm (ε/M.cm^−1^ ± SD × 10^4^, d.i.H_2_O): 204 (16.7 ± 4.75), 244 (5.90 ± 2.34), 290 (5 ± 1.40), 316 (1.94 ± 0.47). IR (cm^−1^): 3418, 3071, 2941, 1609, 1263, 846. CD λ_max_ nm (Δε/M.cm^−1^ × 10^1^, d.i.H_2_O): 214 (−175), 241 (−29.2), 285 (+11.9). ESI-MS: *calculated* for [M-H]^+^: *m*/*z* = 947.25; *experimental*: *m*/*z* = 948.25.

### 2.3. Instrumentation 

A range of equipment was utilised to confirm the homogeneity and composition of the reported platinum(IV) complexes. The protocols performed are summarised in the Appendix A. 

### 2.4. Physicochemical and Biological Investigations

#### 2.4.1. Solubility Measurements

The solubility of complexes **1**–**6** was tested in d.i.H_2_O at room temperature. Small aliquots of d.i.H_2_O were titrated into an Eppendorf tube containing each metal complex (1 mg) until full dissolution. For every titration performed, each sample was vortexed and sonicated. All solubility values were expressed in mg/mL and mol/L.

#### 2.4.2. Stability Studies

The stability measurements for complexes **1**–**6** were determined by dissolving each complex in 10 mM aqueous phosphate buffered saline (PBS) solution at pH ~7.4 and incubated at room temperature and at 37 °C for 36 h. The experiments were monitored via HPLC.

#### 2.4.3. Lipophilicity Studies

Lipophilicity measurements were undertaken using HPLC, as previously described (see Appendix A) [53,55,56,57,58,59].

#### 2.4.4. Reduction Studies

The reduction behaviour of **1**–**6** was probed via ^1^H-NMR and 1D-^195^Pt-NMR spectroscopy, as previously described (see Appendix A) [53,55].

#### 2.4.5. Cell Viability Assays

Cell viability assays were completed at the Calvary Mater Newcastle Hospital, NSW, Australia, as previously reported (see Appendix A) [52,53,55,60]. The GI_50_ values of platinum(II) precursors and platinum(IV) scaffolds, together with cisplatin, oxaliplatin and carboplatin, which were determined using the previously reported method, were also presented in this study for comparison [53,55]. The selectivity cytotoxicity index (SCI) for **1**–**6** and ligands (NPX and ACE) was calculated by dividing the GI_50_ values of the complexes or ligands in the normal breast cell line MCF10A by their GI_50_ in the cancerous prostate cell line Du145. Generally, a greater SCI denotes higher selectivity towards cancer cells [61,62].

#### 2.4.6. Reactive Oxygen Species (ROS) Potential 

To determine the presence of ROS in treated cells, a DCFDA/H_2_DCFDA-cellular ROS Assay Kit (Abcam, Cambridge, MA, USA) was utilised, as previously reported [53,55,63,64]. The ROS potential of platinum(II) precursors and platinum(IV) scaffolds, which were determined using the same method, were also presented in this study for comparison [53].

#### 2.4.7. Mitochondrial Membrane Potential (MtMP)

To study the MtMP changes in treated cells, a TMRE-MtMP Assay Kit (Abcam, Cambridge, MA, USA) was used (see Appendix A) [21].

#### 2.4.8. Cyclooxygenase-2 (COX-2) Inhibition

To determine the levels of COX-2 inhibitory characteristics of **1**–**6**, the COX-2 (human) Inhibitor Screening Assay Kit (Item No. 701080, Cayman Chemical, Ann Arbor, MI, USA) was used (see Appendix A).

## 3. Results and Discussion

### 3.1. Synthesis and Characterisation

The NHS esters of NPX and ACE were synthesised using previously established methods [52,53]. The successful synthesis of the NHS esters was confirmed by HPLC (Appendix A), ^1^H-NMR (Appendix A), and ESI-MS (Appendix A) experiments. The platinum(II) precursors and platinum(IV) scaffolds of types, **[Pt^II^(H_L_)(A_L_)]^2+^** and **[Pt^IV^(H_L_)(A_L_)(OH)_2_]^2+^**, were synthesised as previously described [52,53,55]. To prepare **[Pt^IV^(H_L_)(A_L_)(X)(OH)]^2+^** (**1**–**6**) (Figure 4), previously established protocols were applied [52,53]. All resulting complexes were purified through a flash chromatography system to obtain higher purity. Characterisation confirming the composition and homogeneity of **1**–**6** included HPLC (Appendix A), ^1^H-NMR (Appendix A), 2D-COSY (Appendix A), ^1^H-^195^Pt-HMQC (Appendix A), UV (Appendix A), CD (Appendix A), ESI-MS (Appendix A), and IR (Appendix A). All experimental yields, HPLC peak areas (%) and T_R_, and mass-to-charge ratios (*m*/*z*) of **1**–**6** are outlined in Table 1.

#### 3.1.1. NMR Spectra Assignment

A summary of the ^1^H-NMR and ^1^H-^195^Pt-HMQC data of **1**–**6** is presented in Table 2 and Table 3, including chemical shifts (δ ppm), multiplicity, integration, and calculated *J*-coupling constants (Hz). Due to proton exchange with D_2_O, no amine proton resonances were observed. With respect to the ^1^H-NMR spectrum of **1** (Figure 5), 17 peaks were recorded including the large signal at 4.70 ppm induced by the deuterated solvent used. A notable upfield shift movement was observed along the aromatic region (8–9 ppm) according to the resonances elicited by the phen protons. While this is mostly influenced by the axial coordination of NPX, a more viable explanation to this phenomenon is reflective of the structure of NPX. NPX contains an electron-rich naphthalene group, which may have interacted with the phen ring system of the complex through π–π stacking. Considering the high electron density of the aromatic rings (i.e., phen and naphthalene), this effect generates greater opposition to the applied magnetic field, which causes the hydrocarbons to shield and resonate at a lower frequency.

The protons originating from phen, H2 and H9 resonated as a triplet at 9.12 ppm with a calculated *J*-coupling constant of 5.0 Hz. Due to the deshielding effect induced by the electronegativity of the nitrogen (N) atoms bound to the phen ring, the protons resonated furthest downfield. The two individual doublets at 8.74 and 8.70 ppm were assigned to H4 and H7 protons, respectively, and both have a calculated *J*-coupling constant of 8.3 Hz This was followed by a multiplet at 8.10 ppm, exhibited by H3 and H8 protons. The quartet at 7.67 ppm with a calculated *J*-coupling constant of 8.9 Hz was assigned to protons H5 and H6. Overall, these multiplicities observed for **1** in the aromatic region contrast what is typically reported for its corresponding platinum(II) and platinum(IV) complexes, **PHEN*SS*** and **PHEN*SS*(IV)(OH)_2_**, as they typically resonate as doublets and particularly a singlet for H5 and H6 (Figure 6) [22,52,53,55,56,57,65]. 

As for the protons originating from the naphthalene group of NPX represented by a, b, c, d, e and f, varied multiplicity was also demonstrated (Figure 5). The more deshielded protons near the alkoxy group, c and d, resonated as a multiplet at 7.12 ppm, while the less deshielded protons, a and b, resonated as separate singlets at 7.04 and 6.71 ppm, respectively. Moreover, e resonated as a doublet at 6.84 ppm with a calculated *J*-coupling constant of 8.5 Hz, while f, which is the least deshielded proton in the naphthalene group, resonated at 6.40 ppm as a doublet of doublets with calculated *J*-coupling constants of 1.4 and 8.5 Hz. The sharp singlet at 3.99 ppm was assigned to the methyl at the alkoxy group of NPX represented by γ. The methylene (α) and methyl (β) protons near the carbonyl group of NPX were assigned to the quartet and doublet at 3.50 and 1.15 ppm, respectively. Finally, the chemical multiplicity in the aliphatic region (1–3 ppm) exhibited by the ancillary ligand, *SS*-DACH agrees with the literature data [22,25,52,53,55,56,57,58,59,66].

To further confirm the successful coordination of NPX to the platinum core and only occupied one axial position, ^1^H-^195^Pt-HMQC experiments were undertaken at –2800 and 400 ppm. Normally, the platinum(II) scaffolds used in this study resonate at −2800 ppm while their corresponding platinum(IV) derivatives resonate at 400 ppm, as previously reported [22,25,52,53,55,56,57,58,59,66]. According to the ^1^H-^195^Pt-HMQC spectrum of **1** (Figure 7), the correlation of the phen protons to the platinum was confirmed when two peaks resonated at 545 ppm. Specifically, the cross-coupling of H2 and H9 (9.12 ppm) and of H3 and H8 (8.10 ppm) protons with the platinum peaks at 545 ppm demonstrate correlation. 

With respect to the ^1^H-NMR spectrum of **2** (Appendix A), a few differences in multiplicity were noted compared to the results observed for **1** (Figure 5) and **3** (Appendix A), specifically the multiplicity in the aromatic region (8–9 ppm). For example, the H2 and H9 protons of **2** appeared as two separate signals, a triplet at 9.12 ppm and a doublet 9.04 ppm, exhibiting the same *J*-coupling constant of 5.6 Hz, respectively. Additionally, the two separate quartets at 8.84 and 8.60 ppm were assigned to H4 and H7 protons, while the multiplet at 8.10 ppm was assigned to H3 and H8 protons. The H6 proton resonated as a doublet at 7.40 ppm, with a *J*-coupling constant of 19 Hz. This high *J*-coupling constant is a result of the long-range coupling between H6 and its neighbouring methyl group at the 5 position (consisting of three protons) in the heterocyclic ring system (Appendix A). Overall, these differences in multiplicity of the heterocyclic protons are indicative of the structure of the complexes, considering that **2** and **3** contain methyl groups in their heterocyclic ring systems, while **1** does not. Nonetheless, the successful coordination of NPX to platinum was confirmed according to the ^1^H-^195^Pt-HMQC results observed for **2** (Appendix A) and **3** (Appendix A). 

Furthermore, the results obtained for the remaining complexes incorporating the ACE ligand, **4**–**6**, also showed distinct differences in their chemical shifts, which are attributed to the axial coordination of ACE and the number of methyl groups in the heterocyclic ligands. The assignment of resonances for the ^1^H-NMR spectra of **4**–**6** (Table 3 and Appendix A) followed the same rationale to that described for **1**–**3**. Lastly, the successful coordination of ACE to platinum at one axial position was also confirmed by ^1^H-^195^Pt-HMQC (Appendix A).

#### 3.1.2. Electronic Spectra Analysis

The electronic transitions demonstrated in the UV measurements were comparable with similar complexes in the literature [22,52,53,55,56,57,58,59,66]. The UV spectra of **1**–**6** (Appendix A) were acquired as previously reported [52,53,55]. In addition, CD experiments were undertaken to confirm if the chirality of the ancillary ligand, *SS*-DACH, was retained by the resulting complexes. The patterns observed in the CD spectra **1**–**6** (Appendix A) were also consistent with the literature data [22,52,53,55,56,57,58,59,66]. All notable peaks in the UV and CD spectra of the complexes are outlined in Table 4.

The UV spectra for **1**–**6** (Appendix A) were recorded. For the UV measurements, both metal-to-ligand charge transfer interactions and π–π* transitions were observed. These are primarily influenced by the heterocyclic ligands of the complexes (i.e., phen, 5-Mephen, and 5,6-Me_2_phen) that demonstrate ligand-centred π–π* transitions. The UV spectra of platinum(IV) derivatives incorporating NPX (**1**–**3**) exhibited similar absorption bands (Appendix A and Figure 8). Three prominent absorption bands were recorded at the wavelengths, ~200, ~230 and ~279–289 nm. The differences in the methylation of the heterocyclic ligands of the complexes resulted in slight bathochromic shifts or red shifts, as shown in Figure 8. This pattern is also in agreement with the literature data [25,52,53,55,56,57,58,59,66,67]. Of further note, a prominent absorption band at ~235 nm was acquired for the uncoordinated NPX ligand, followed by three small peaks between 260 and 290 nm. Upon coordination of NPX to platinum(IV), the band at ~235 nm was shifted to lower wavelengths (Figure 8). This may be influenced by the π–π interactions between the aromatic system of NPX and the heterocyclic system of the platinum(IV), since larger conjugated systems tend to absorb at lower wavelengths. 

In comparison, the UV spectra of the platinum(IV) derivatives incorporating ACE (**4**–**6**) also followed a similar trend to **1**–**3**, particularly the red shift recorded between 270 and 290 nm (Figure 9 and Appendix A). The absorption bands recorded for **4**–**6** were slightly blue shifted compared to the absorption bands measured for **1**–**3**. This may be due to the structural differences of the axial ligands, NPX and ACE. NPX contains a naphthalene group that experiences hyperconjugation, which may be increased by the presence of the oxygen (O) in its alkoxy group. The O at the alkoxy group is electron-donating, thus this increases the delocalisation of electrons within the naphthalene ring. On the contrary, while ACE also contains aromatic rings, these are surrounded by electron-withdrawing groups, such as Cl and N atoms, and a carbonyl that reduces electron density.

Since **1**–**6** incorporate a chiral ancillary ligand, *SS*-DACH, it was only appropriate to confirm if the complexes’ chirality has been retained during synthesis. Chirality is an essential parameter that influences the potency of the complexes, and it has been previously established that by substituting the *SS*-DACH to its enantiomer, *RR*-DACH, significant differences in overall cytotoxicity of the complexes were observed [17,18,20,65], where those complexes incorporating *RR*-DACH were significantly less potent than those complexes that incorporate the *SS*-DACH as the ancillary ligand. 

For the CD measurements of **1**–**6**, variations were exhibited (Appendix A). The absorption bands of **1**–**3** (Appendix A) were more defined and prominent compared to the absorption bands of **4**–**6** (Appendix A). Evidently, **1**–**3** displayed stronger positive and negative absorption bands, while **4**–**6** displayed weaker positive and negative absorption bands (Table 4). Prominent positive absorption bands at the lower wavelengths (239–244 nm) were exhibited by **1**–**3** (Appendix A), which were not recorded for **4**–**6** (Appendix A). These variations are likely more influenced by the axial ligands, NPX and ACE, rather than the methylation of the heterocyclic systems of the complexes. In summary, except for the strong positive absorption bands exhibited by **1**–**3** between 239–244 nm, the CD measurements obtained follow a comparable trend with published examples of platinum(IV) complexes [22,52,53,55,56,57,58,59,66], confirming that chirality was retained during synthesis. 

IR measurements were also undertaken to further verify the functional groups present in the structures of **1**–**6** (Appendix A). Some of the prominent IR absorption peaks recorded for the complexes are summarised in Table 5. Because of the structural similarities of the complexes, the peaks for all complexes follow almost the same pattern (Appendix A). As shown in Table 5, the types of bonds observed in the IR spectra of **1**–**6** included the following: O-H stretch (3410–3385 cm^−1^), C-H aromatic stretch (3062–3076 cm^−1^), C-H alkyl stretch (2936–2943 cm^−1^), C-C aromatic stretch (1603–1609 cm^−1^), C-O alkyl aryl ether stretch (1218–1265 cm^−1^) and a C-Cl halogen stretch (845–846 cm^−1^). These functional groups are also present in the complexes. The C-Cl stretch was only exhibited by the platinum(IV) complexes incorporating the ACE axial ligand, **4**–**6** (845–846 cm^−1^), considering that ACE is the only axial ligand that has a Cl atom. Because of the multiple functional groups present in the complexes, the frequencies observed cannot be assigned to a specific functional group. For example, the C-H and C-C vibrations from the heterocyclic ligands (phen, 5-Mephen and 5,6-Me_2_phen) of the platinum(II) cores may exhibit overlapping vibrations as those with the coordinated ACE and NPX ligands. Overall, the vibrations observed in the IR spectra of **1**–**6** are comparable to literature data [68,69,70,71,72].

#### 3.1.3. Solubility Measurements

Solubility is an essential parameter that can predict drug effectiveness [73]. Generally, a drug with poor solubility will only promote pharmacokinetic challenges (Figure 10). Platinum(II) drugs such as cisplatin, oxaliplatin and carboplatin are effective; however, they cannot be administered orally due to their limited solubility in aqueous conditions [36]. The poor solubility of these drugs is also a consequence of their poor bioavailability, and generally, high-dose administration is often required to achieve adequate pharmacological response. Although, high-dose administration can increase patient burden, especially the risks of patients developing adverse side effects considering the high toxicity of the drugs [74]. 

The inherent characteristics of platinum(IV) complexes have inspired optimism that an effective orally available prodrug can be designed. Evidence has been demonstrated previously with *bis*-(acetate)-ammine dichloro-(cyclohexylamine) platinum(IV) (satraplatin), which was the first oral platinum agent that took part in phase III clinical trials but had to be discontinued due to low overall survival benefit [75]. The availability of orally available anticancer prodrugs will undoubtedly improve treatment experience and patient care, as hospitalisation would not be necessary.

In this study, the solubility for **1**–**6** in d.i.H_2_O at room temperature was determined. All values were expressed in mg/mL and mol/L, as shown in Table 6. Complexes **1**–**6** were found to be more soluble than cisplatin, oxaliplatin and carboplatin in water (Table 6). The results indicate that the studied complexes should have better bioavailability than clinically used platinum(II) drugs and may be suited for oral administration. 

#### 3.1.4. Lipophilicity Measurements

In conjunction with solubility, lipophilicity also plays a vital role in drug effectiveness, as it directly impacts the diffusion of a drug through cell membranes (or permeability) [77]. Generally, NSAIDs (i.e., NPX and ACE) are highly lipophilic substances because they contain lipophilic groups such as carboxylic acids and aromatic rings [78]. Moreover, the biological activity of NSAIDs is mostly dependent on lipophilicity [79]. Since complexes **1**–**6** incorporate the NSAIDs, NPX and ACE, it was only appropriate to determine their lipophilicity.

Lipophilicity measurements were undertaken using HPLC as previously described [53,55,56,57,58,59]. The complexes were eluted at different isocratic ratios. The capacity factor (k) was calculated using the retention times of the complexes. A standard curve was generated to determine the log value of the capacity factor (log k), and this was performed by plotting the log k value at each isocratic ratio versus the percentage of the organic solvent used in solvent (Appendix A). This produced a linear expression that allowed for the extrapolation of log k_w_. Log k_w_ is the measure of lipophilicity; therefore, a greater value corresponds to increased lipophilicity. 

The rank of complexes by increasing lipophilicity is **1** < **2** < **3** < **4** < **5** < **6** (Table 7). It is evident that those complexes containing the smaller axial ligand NPX (**1**–**3**) are less lipophilic than those derivatives containing the larger axial ligand ACE (**4**–**6**). This trend also corresponds to the reported lipophilicity values of NPX (2.88) and ACE (4.49) [80]. Moreover, the methylation at the heterocyclic systems of the complexes also influenced lipophilicity, since the 5,6-Me_2_phen derivatives (**3** and **6**) were more lipophilic than the phen derivatives (**1** and **4**), and this observation is also comparable with the literature data [53,55,56,57,58,59]. Overall, the axial coordinated NPX and ACE ligands had more influence on the overall lipophilicity of the complexes than the change in methylation in their heterocyclic ligands.

#### 3.1.5. Preliminary Reduction Studies 

Platinum(IV) complexes are expected to maintain their octahedral structure in the bloodstream prior to their activation inside cancer cells [81]. The intracellular activation of platinum(IV) complexes is typically initiated by biological reducing agents such as AsA or GSH, which act as catalysts in the reduction of platinum(IV) complexes to their corresponding active platinum(II) species, along with the release of the axial ligands [34,81,82,83]. Because of this phenomenon, platinum(IV) complexes are considered prodrugs.

The reduction behaviour of **1**–**6** was monitored by ^1^H-NMR and 1D-^195^Pt-NMR spectroscopy using previously established methods [53,55]. Prior to the reduction measurements, each metal complex was dissolved with PBS in D_2_O only. Initial ^1^H-NMR and 1D-^195^Pt-NMR experiments measured at 400 and −2800 ppm (30 min per region) at 37 °C were performed to confirm the purity of the complexes, and most importantly, to show that the PBS does not affect the structural integrity of the complexes and causes reduction.

Based on the initial 1D-^195^Pt-NMR spectra acquired (Appendix A), the structural integrity of the complexes was retained, and no signs of reduction were observed, thus also confirming that the PBS had no effect on the complexes. Subsequently, AsA was added in each solution containing the metal complex, PBS and D_2_O. Then, sequential ^1^H-NMR experiments were undertaken for 1 h, followed by final 1D-^195^Pt-NMR experiments measured within the regions of 400 and −2800 ppm (Appendix A). The approximate time points for **1**–**6** at which an estimated 50 and 100% reduction had occurred, in the presence of AsA, as represented by T_50%_ and T_100%_, are summarised in Table 8. 

In summary, the platinum(IV)-NPX derivatives (**1**–**3**) reduced gradually in the presence of AsA compared to the platinum(IV)-ACE derivatives (**4**–**6**), which reduced rapidly (Table 8). The differences in the reduction times are more influenced by the axial ligands rather than by the methylation at the heterocyclic ligands of the complexes. The ^1^H-NMR reduction spectra obtained for **1** demonstrated notable movement of resonances originating from the protons of the heterocyclic ligand (phen) and the NPX ligand (Figure 11). The resonances of H2 and H9 protons had shifted upfield and overlapped with the resonances of H4 and H7 protons, as shown in Figure 11. Moreover, the resonances of H3 and H8 protons shifted upfield, while the resonances of H5 and H6 protons shifted downfield (Figure 11). Finally, small peaks (with slight overlapping) of the uncoordinated NPX ligand were also recorded between 7.2 and 7.7 ppm, as shown in Figure 11. 

Immediately after the ^1^H-NMR experiments, 1D-^195^Pt-NMR experiments were measured on the regions 400 and −2800 ppm. After 40 min from the final ^1^H-NMR experiment, there was still a platinum peak detected in the 1D-^195^Pt-NMR spectrum of **1** at 400 ppm, as shown in Figure 12. A more prominent platinum peak was recorded at −2800 ppm 30 min after the preceding 1D-^195^Pt-NMR experiment, which indicates the presence of the corresponding platinum(II) precursor, **PHEN*SS*** (Figure 12). The results suggest that complex **1** is a potential prodrug of **PHEN*SS*** because it can successfully reduce back to its platinum(II) precursor. 

Furthermore, the changes in chemical shifts observed for **1** were also comparable in the ^1^H-NMR and 1D-^195^Pt-NMR reduction spectra of complexes **2** and **3** (Appendix A). The only difference is that **2** and **3** had completely reduced to their corresponding platinum(II) precursors, **5ME*SS*** and **56ME*SS***, respectively, at ~1.5 h based on the final 1D-^195^Pt-NMR experiment that was measured at 400 ppm, due to the absence of platinum(IV) peaks (Appendix A). Additionally, the platinum(IV)-ACE derivatives (**4**–**6**) reduced rapidly in the presence of AsA, which only occurred within 10 min according to the acquired ^1^H-NMR spectra (Appendix A).

During the experiments, it was observed that upon adding and mixing AsA with the sample solutions of these complexes (**4**, **5** and **6**), the solutions turned cloudy within 30 s. For these complexes, T_50%_ was within 5 min, as shown in Table 8. Complex **6** had completely reduced at 5 min, while for **4** and **5**, full reduction was achieved between 10 and 15 min (Table 8). Overall, the results indicate that **1**–**6** can successfully reduce to their corresponding platinum(II) precursors; thus, it is correct to conclude that they are prodrugs. 

#### 3.1.6. Stability Studies

Additional stability studies for **1**–**6** were undertaken using 10 mM PBS (~7.4 pH) as the aqueous solution, but without the presence of a reducing agent such as AsA. This was only to show whether the complexes would be stable in aqueous solution for long hours. Each complex was dissolved in PBS and incubated at room temperature and at 37 °C for 36 h. After the incubation, all samples were analysed through HPLC. The platinum(IV) derivatives incorporating NPX (**1**–**3**) were relatively stable for 36 h in PBS at room temperature and at 37 °C, as shown in Appendix A. Although for **2**, some traces of precursor platinum(IV) complexes and the dissociated NPX ligand were evident (Appendix A). In contrast, the platinum(IV) derivatives incorporating ACE (**4**–**6**) were less stable for 36 h in PBS at room temperature and at 37 °C (Appendix A), compared with **1**–**3**. After 36 h at room temperature, peak traces of platinum(IV) complexes and the platinum(II) precursors, as well as the dissociated ACE ligand, were present, as shown in Appendix A. Treatment of **4**–**6** at 37 °C for 36 h resulted in a major reduction of the complexes (Appendix A). From this, it was reasoned that the larger the ligand conjugated to our platinum(IV) scaffold (i.e., ACE), the more susceptible the resulting derivatives (**4**–**6**) are to reduction, as they reduce even without a reducing agent. This parallels the results from the reduction experiments above, where **4**–**6** were the quickest to reduce in the presence of AsA. Overall, the stability of the complexes was more influenced by the axial ligands, NPX and ACE, more so than the methylation in their heterocyclic ligands. 

### 3.2. Biological Investigations

#### 3.2.1. Growth Inhibition Studies

The reported platinum(IV) complexes (**1**–**6**) and their axial ligands, NPX and ACE, were evaluated for antiproliferative activity in twelve cell lines including HT29 colon, U87 glioblastoma, MCF7 breast, A2780 ovarian, H460 lung, A431 skin, Du145 prostate, BE2C neuroblastoma, SJG2 glioblastoma, MIA pancreas, ADDP ovarian variant, and the non-tumour-derived MCF10A breast line. Since current platinum(II) drugs are generally used to treat genitourinary cancers such as ovarian cancer, A2780 ovarian and the resistant ADDP ovarian cell lines were selected to probe the capacity of **1**–**6** to reverse and overcome chemoresistance. Compound growth inhibition was assessed using the MTT assay after 72 h of treatment. All determined GI_50_ values are summarised in Table 9, including the GI_50_ values of the platinum(II) precursors and platinum(IV) scaffolds, as well as cisplatin, oxaliplatin and carboplatin [53,55], to allow for comparison.

The platinum(II) precursors (**PHEN*SS***, **5ME*SS*** and **56ME*SS***) and platinum(IV) scaffolds (**PHEN*SS*(IV)(OH)_2_**, **5ME*SS*(IV)(OH)_2_** and **56ME*SS*(IV)(OH)_2_**) utilised in this study to create the studied complexes, **1**–**6**, display exceptional potency, substantially greater than cisplatin, oxaliplatin and carboplatin (Table 9). It was our interest to further enhance the biological activity of these complexes by conjugating the NSAIDs, NPX and ACE, as axial ligands to their cores. Notably, GI_50_ values ranging between 0.22 and 2570 nM were recorded for **1**–**6**, as summarised in Table 9. This indicates their superiority as anticancer agents, especially when compared with cisplatin, oxaliplatin and carboplatin. 

The calculated mean GI_50_ values for **1**–**6** in the entire cell line tested was 32–931 nM, which is significantly lower than the calculated mean GI_50_ values for cisplatin, oxaliplatin and carboplatin (1463–32,242 nM) (Table 9). The platinum(IV) derivatives incorporating the heterocyclic ligands, 5-Mephen (**2** and **5**) and 5,6-Me_2_phen (**3** and **6**) were more growth inhibitory than those incorporating the phen ligand (**1** and **4**), as expected (Table 9). The trend observed here confirms once again that the methylation of the heterocyclic ligand influences biological activity, and this has been translated in previous studies [16,24,53,55,56,57,58,59,66]. Furthermore, the platinum(IV) derivatives incorporating the NPX ligand (**1**–**3**) were less potent than those platinum(IV) derivatives incorporating the ACE ligand (**4**–**6**). This is likely due to the response of NPX and ACE intracellularly, despite having the same GI_50_ values of >50,000 nM (Table 9). Nonetheless, this may also suggest that the ACE ligand has more synergism with platinum than the NPX ligand.

Relative to the corresponding platinum(IV) scaffolds of **1**–**6** (**PHEN*SS*(IV)(OH)_2_**, **5ME*SS*(IV)(OH)_2_** and **56ME*SS*(IV)(OH)_2_**), enhancement in cytotoxicity was observed upon coordination of NPX and ACE (Table 9). For example, in the MCF7 breast cell line, **PHEN*SS*(IV)(OH)_2_** had a GI_50_ value of 16,000 ± 4500 nM, which decreased by 14- and 19-fold upon coordination of NPX and ACE to create **1** (1120 ± 342 nM) and **4** (840 ± 390 nM), respectively. **5ME*SS*(IV)(OH)_2_** elicited a GI_50_ value of 1400 ± 300 nM in the BE2C neuroblastoma cell line, to which this also decreased by 4- and 5.6-fold when NPX and ACE were conjugated to its core to create **2** (680 ± 17 nM) and **5** (250 ± 33 nM), respectively. Moreover, in the H460 lung cell line, **56ME*SS*(IV)(OH)_2_** (190 ± 150 nM) was 3- and almost 12-fold less potent than its derivatives, **3** (56 ± 5 nM) and **6** (16 ± 2 nM), respectively.

When comparing the in vitro cytotoxicity of **1**–**6** with their corresponding platinum(II) precursor complexes (**PHEN*SS***, **5ME*SS*** and **56ME*SS***), notable differences were also observed. **PHEN*SS*** and **5ME*SS*** were more potent than **1**, **2** and **4**, which are their platinum(IV) derivatives. The phen derivatives, **1** and **4**, had calculated mean GI_50_ values of 931 ± 139 and 735 ± 58 nM, respectively, demonstrating a 1- to 2-fold difference compared to the mean GI_50_ value of **PHEN*SS*** (434 ± 110 nM) (Table 9). The calculated mean GI_50_ value of **2** (490 ± 58 nM) is 4-fold more than the mean GI_50_ value of **5ME*SS*** (117 ± 12 nM). However, derivative (**5**) had a calculated mean GI_50_ value of 70 ± 12 nM over all cell lines, which infers that it is more potent than its platinum(II) precursor, **5ME*SS*** (Table 9). Our lead complex, **56ME*SS***, had a calculated mean GI_50_ value of 36 ± 10 nM over the entire range of cell lines, which proves to be more potent than its derivative, **3**. Remarkably, the other derivative (**6**) proved to be marginally more potent than **56ME*SS***, eliciting the lowest GI_50_ values (Table 9).

The most biologically active platinum(IV) derivatives in the study were **5** and **6**, which incorporated the 5-Mephen and 5,6-Me_2_phen heterocyclic ligands, respectively, and ACE as the axial ligand. Compound **5** elicited a GI_50_ value of 1.3 ± 0.4 nM in the Du145 prostate cell line, which makes it 17-fold more potent than its platinum(II) precursor, **5ME*SS*** (22 ± 3 nM), at least 920-fold more potent than cisplatin (1200 ± 100 nM), and 2230-fold and 11,540-fold more potent than oxaliplatin (2900 ± 400 nM) and carboplatin (15,000 ± 100 nM), respectively (Table 9). Additionally, **5** also responded strongly in the ADDP ovarian variant cell line, eliciting a GI_50_ value of 9.3 ± 2 nM, making it 3000-fold more potent than cisplatin (28,000 ± 1600 nM) in that cell line. Finally, **6** proved to be more potent than its platinum(II) precursor **56ME*SS*** in all cell lines, except for the BE2C neuroblastoma and SJG2 glioblastoma cell lines (Table 9). The in vitro cytotoxicity of **6** was significantly enhanced in nine cell lines: HT29 colon (4.7 ± 4 nM), MCF7 breast (12 ± 6 nM), A2780 ovarian (10 ± 4 nM), H460 lung (16 ± 2 nM), A431 skin (16 ± 12 nM), Du145 prostate (0.22 ± 0.1 nM), MIA pancreas (7.8 ± 4 nM), MCF10A breast (3.4 ± 1 nM), and the ADDP ovarian variant (1.3 ± 0.5 nM) (Table 9). Notably, in the Du145 prostate cell line, **6** proved to be almost 21-fold potent than **56ME*SS*** (4.6 ± 0.4 nM) and at least 5450-fold more potent than cisplatin (1200 ± 100 nM) in the same cell line, as shown in Table 9. Moreover, **6** was more potent in the cisplatin-resistant ADDP ovarian cells (1.3 ± 0.5 nM) than in the parental A2780 cisplatin-sensitive ovarian cells (10 ± 4 nM). Collectively, these outcomes suggest that **1**–**6** are not sensitive to the drug resistance mechanisms observed from typical cisplatin-based regimens, as indicated by their potency in the ADDP ovarian variant cell line. 

The selectivity cytotoxicity index (SCI) for the metal complexes and compounds (NPX and ACE) was also calculated by dividing the GI_50_ values of the complexes or compounds in MCF10A (non-tumour) by their GI_50_ in Du145, which is reported in Table 9. The SCI gives a measure of the efficacy of the drug against cancer cells: the greater the SCI the higher the selectivity towards cancer cells [61,62]. The high SCI exhibited by **5** and **6** towards the Du145 cell line of 15.4 and 15.5 are notable (Table 9). In contrast, the platinum(II) precursors (**PHEN*SS***, **5ME*SS*** and **56ME*SS***), platinum(IV) scaffolds (**PHEN*SS*(IV)(OH)_2_**, **5ME*SS*(IV)(OH)_2_** and **56ME*SS*(IV)(OH)_2_**), complexes **1**–**4**, cisplatin, carboplatin, NPX and ACE showed little selectivity towards Du145, with SCI values ranging between 1.00 and 5.48. From this, it can be inferred that the coordination of the ACE ligand to the platinum(IV) scaffolds **5ME*SS*(IV)(OH)_2_** and **56ME*SS*(IV)(OH)_2_** had significantly improved the selectivity of the resultant platinum(IV) derivatives **5** and **6** towards prostate cancer cell populations.

Furthermore, platinum(IV) complexes utilising the cores of platinum(II) drugs incorporating NPX as axial ligands have also been investigated for their anticancer properties, most specifically in the MCF7 cell line [37,72]. Tolan et al. synthesised cisplatin-NPX, oxaliplatin-NPX and carboplatin-NPX complexes, which demonstrated GI_50_ values of 10,400 ± 790, 9470 ± 750 and 9120 ± 630 nM, respectively, in the MCF7 cell line [72]. These values are still substantially higher than the GI_50_ values elicited by our platinum(IV) complexes containing NPX (**1**–**3**) in that cell line (Table 9). For example, **3** (60 ± 0.5 nM) proved to be 173-fold more potent than cisplatin-NPX (10,400 ± 790 nM), 158-fold more potent than oxaliplatin-NPX (9470 ± 750 nM), and 152-fold more potent than carboplatin-NPX (9120 ± 630 nM) in the MCF7 cell line. It is only appropriate to mention that these differences in cytotoxicity are mostly attributed to the biological activity of the platinum(II) cores of these complexes, considering that our platinum(II) scaffolds (**PHEN*SS***, **5ME*SS*** and **56ME*SS***) are significantly more potent than cisplatin, oxaliplatin and carboplatin.

#### 3.2.2. ROS Potential

ROS is a family of highly reactive chemicals generated in the mitochondria, which are also natural by-products of cellular metabolic processes [84]. The regulation of ROS is critical for normal biological functions; however, if compromised by stress factors (i.e., chemotherapy drugs), it can lead to oxidative stress [85]. Oxidative stress reflects the imbalance between ROS production and the ability of a biological system to detoxify these. Clearly, the abundant accumulation of ROS has deleterious side effects on DNA, proteins, lipids, and other cellular components that are vital for life. While this phenomenon is detrimental to normal healthy cells, this can be applied as a treatment strategy against cancer cells, as conveyed in the literature [86,87,88,89]. Most importantly, there are two roles of ROS activity in cancer. With the right amount of ROS intracellularly, it can promote metastasis and help cancer cells acquire resistance to treatment [90]. On the contrary, if the ROS levels are significantly elevated and prolonged, cytotoxic effects would transpire and therefore induce apoptosis and potentially inhibit the resistance mechanisms of cancer cells to treatment [91].

In this study, the ROS activity of **1**–**6**, together with their axial ligands (NPX and ACE), and cisplatin in the human colon cancer cell line, HT29 at 24, 48 and 72 h are reported (Table 10, Figure 13 and Appendix A). A summary of measured fluorescence in RFU units for the complexes and ligands is summarised in Table 10. Additionally, the ROS potential of the platinum(II) precursors and platinum(IV) scaffolds was determined previously [53,55] and was presented here to allow for comparison (Table 10 and Figure 13). The HT29 colon cell line was selected for testing, as it is one of the cell lines wherein **1**–**6** demonstrated exceptional in vitro cytotoxicity (Table 9). Treatment with the complexes and ligands was completed at each specific GI_50_ concentration.

A significant increase in ROS production was detected for **1**–**6** and the axial ligands, NPX and ACE, in HT29 colon cells at 24 and 48 h, relative to the control (Table 10 and Figure 13). This increase also corresponds to the ROS activity exhibited by the platinum(II) precursors and platinum(IV) scaffolds, and cisplatin, as shown in Table 10 and Figure 13. Notably, the RFU values of **1**–**6** were higher compared with the RFU values of their corresponding platinum(II) precursors and platinum(IV) scaffolds at 24 and 48 h (Table 10 and Figure 13). This confirms that the conjugation of NPX and ACE to our platinum(IV) scaffolds enhanced ROS production. A consistent decline in ROS activity was demonstrated by **1**–**6** and the NPX and ACE ligands, particularly the drastic decreased ROS activity at 72 h (Table 10 and Figure 13). For the platinum(IV) derivatives incorporating NPX, **1**–**3**, their RFU values at 72 h (**1**: 63 RFU; **2**: 84 RFU; **3**: 104 RFU) were lower than the RFU value of the control (131 RFU) (Table 10 and Figure 13). While this trend also paralleled the ROS activity of the platinum(IV) derivatives incorporating ACE, **4**–**6**, their RFU values at 72 h (**4**: 168 RFU; **5**: 192 RFU; **6**: 202 RFU) were still higher than the control (131 RFU) (Table 10 and Figure 13). Nonetheless, this consistent decline in ROS production observed for **1**–**6** and NPX and ACE, was not demonstrated by cisplatin, platinum(II) precursors or platinum(IV) scaffolds, since the ROS activity of these complexes remained significantly high at 24, 48 and 72 h, relative to the control (Table 10). 

We conclude that the initial increase in ROS production at 24 h, followed by the decline in ROS activity at 48 and 72 h, exhibited by **1**–**6**, is reflective of the pro-oxidant and antioxidant effects of the coordinated axial ligands (or NSAIDs), NPX and ACE, intracellularly towards ROS. There is sufficient evidence suggesting that NSAIDs can both effectively induce ROS production and scavenge ROS [92,93,94,95]. The results acquired from this study indicate that the studied platinum(IV) complexes, **1**–**6**, can promote cellular damage towards cancer cells by production of ROS. 

#### 3.2.3. Mitochondrial Membrane Potential (MtMP)

Mitochondria is the powerhouse of the cell, and most significantly, the main driver in creating energy for metabolic processes to sustain normal cellular functions [96]. Exogenous stressors such as chemotherapy drugs (and other metallodrugs) are known to induce bioenergetic stress in the mitochondria and cause MtMP disruption, which then activates the mitochondrial apoptotic pathway [21,97,98,99,100,101]. MtMP is a critical predictor of mitochondrial activity that reflects the electrical potential difference between the intracellular and extracellular environment of a cell [102]. Loss in MtMP is indicative of bioenergetic stress that may result in apoptosis.

Here, we report the MtMP changes upon treatment of **1**–**6**, NPX, ACE, the platinum(II) precursors (**PHEN*SS***, **5ME*SS*** and **56ME*SS***), platinum(IV) scaffolds (**PHEN*SS*(IV)(OH)_2_**, **5ME*SS*(IV)(OH)_2_** and **56ME*SS*(IV)(OH)_2_**), and cisplatin in HT29 colon cells at 24, 48 and 72 h (Table 11 and Figure 14). A summary of measured fluorescence in RFU units for the complexes and ligands is summarised in Table 11. TMRE was used to stain HT29 colon cells, and treatment with the complexes and ligands was completed at each specific GI_50_ concentration. The MtMP changes detected are expressed as RFU. Additionally, lower RFU values correspond to a decline in mitochondrial activity. 

From the results, a decrease in mitochondrial activity was detected upon treatment of cisplatin, the platinum(II) precursors (**PHEN*SS***, **5ME*SS*** and **56ME*SS***) and platinum(IV) scaffolds (**PHEN*SS*(IV)(OH)_2_**, **5ME*SS*(IV)(OH)_2_** and **56ME*SS*(IV)(OH)_2_**), the studied platinum(IV) complexes, **1**–**6**, and the axial ligands, NPX and ACE, up to 72 h (Table 11 and Figure 14). Particularly for **1**–**6**, the decrease in mitochondrial activity for the duration of the treatment followed the same trend that was observed in the ROS activity detected for these complexes (Table 10, Figure 13 and Appendix A). This may suggest that the decline in mitochondrial activity is due to the accumulation of ROS induced by the complexes, considering that increased ROS production can impair mitochondrial functions and progressive cell dysfunction. This observation is also proof that there is a correlation between ROS production and MtMP changes (Figure 13 and Figure 14) [98,103,104]. The findings indicated the ability of **1**–**6** to cause MtMP changes, by reducing the mitochondrial activity up to 72 h in HT29 colon cells. Although, it is noteworthy that the MtMP changes detected for the complexes were mostly influenced by their corresponding platinum(II) precursors (**PHEN*SS***, **5ME*SS*** and **56ME*SS***) rather than by the released axial ligands, NPX and ACE. 

#### 3.2.4. Cyclooxygenase-2 (COX) Inhibition

NSAIDs are well recognised as inhibitors of the COX enzymes, COX-1 and COX-2 [37,42,45]. While both COX-1 and COX-2 are responsible in prostaglandin production, they are distinct from one another in terms of their tissue distribution and regulation of gene expression [49,105]. COX-1 is typically expressed in most tissues and is associated with normal physiological functions, particularly by maintaining prostaglandin production at basal levels [49,50]. In contrast, COX-2 is involved in the excessive production of prostaglandin during inflammation (or hypoxia), and extensive studies have shown that COX-2 resultant prostaglandins are overexpressed in tumours [49,50]. Most importantly, the overexpression of COX-2 promotes tumour development and progression [51,106]; therefore, COX-2 is a more valuable target than COX-1. To explore the COX-2 inhibitory properties of the studied complexes (**1**–**6**), ELISA was used against the COX-2 enzyme. Here, we report the % of COX-2 inhibitory activity of **1**–**6**, NPX, ACE, the platinum(II) precursors (**PHEN*SS***, **5ME*SS*** and **56ME*SS***) and cisplatin at 72 h (Table 12, Figure 15 and Appendix A). A summary of the measured COX-2 percentage inhibition for the complexes and ligands is outlined in Table 12.

The greatest COX-2 inhibitory activity was demonstrated by the COX-inhibiting ligands, NPX (83%) and ACE (76%) (Table 12 and Figure 15). In contrast, the platinum(II) precursors (**PHEN*SS***, **5ME*SS*** and **56ME*SS***) and cisplatin showed the lowest inhibitory response towards COX-2 (Table 12 and Figure 15), with the COX-2 inhibition ranging from only 5–9%. Upon coordination of NPX and ACE to our platinum(IV) scaffolds, the COX-2 inhibition for **1**–**6** improved significantly, when compared with the COX-2 inhibition of **PHEN*SS***, **5ME*SS***, **56ME*SS*** and cisplatin (Table 12 and Figure 15). However, relative to the significant COX-2 inhibition demonstrated by NPX (83%) and ACE (76%), the COX-2 inhibitory activity of **1**–**6** was comparatively lower (Table 12 and Figure 15). This may be dependent on the reductive state of the complexes **1**–**6** in vitro and the ability to release COX-inhibiting ligands. Furthermore, the platinum(IV) derivatives of **PHEN*SS*** and **5ME*SS*** (**1**, **2**, **4** and **5**) elicited more inhibitory activity towards COX-2 compared to the platinum(IV) derivatives of **56ME*SS*** (**3** and **6**). It is also evident that COX-2 inhibition of the platinum(IV)-NPX derivatives (**1**–**3**) was slightly greater than for those derivatives containing ACE (**4**–**6**) (Table 12 and Figure 15). The results suggest that COX-2 inhibition is independent of the cytotoxicity and lipophilicity of the complexes. The most cytotoxic and lipophilic complexes contain the ACE ligand (**4**–**6**), which exhibited a reduced response towards inhibiting COX-2, while the least cytotoxic and lipophilic complexes incorporating NPX (**1**–**3**) exhibited a marginally improved response towards inhibiting COX-2. These findings confirm that the coordination of the NSAIDs, NPX and ACE, to our platinum(IV) scaffolds generated complexes that are capable of inhibiting COX-2, may potentially be effective in reducing tumour-related inflammation, and may also show selectivity towards tumours that overexpress COX-2 resultant prostaglandins.

## 4. Conclusions

Six mono-substituted platinum(IV) complexes (**1**–**6**) incorporating the NSAIDs, NPX and ACE, were synthesised, characterised and evaluated for their antitumour activity. A range of spectroscopic and spectrometric techniques confirmed the composition and homogeneity of the complexes. Solubility tests were performed at room temperature in d.i.H_2_O. Complexes **1**–**6** were found to be more soluble than cisplatin, oxaliplatin and carboplatin. ^1^H-NMR and 1D-^195^Pt-NMR provided insight on the reduction properties of the complexes in PBS (~7.4 pH) and AsA at 37 °C. Complexes **1**–**6** were reduced to their corresponding platinum(II) precursors and released the axial ligands, indicating their potential as prodrugs. Those derivatives incorporating the smaller ligand NPX (**1**–**3**) reduced slower, while those derivatives incorporating the larger ligand ACE (**4**–**6**) reduced rapidly in the presence of AsA. Stability and lipophilicity measurements were also undertaken using HPLC. Complexes **1**–**3** were stable in PBS at room temperature and at 37 °C for 36 h, unlike **4**–**6** which exhibited major reduction. According to the log k_w_ values, **4**–**6** proved to be more lipophilic than **1**–**3**. 

The in vitro cytotoxicity of **1**–**6** in multiple cell lines was assessed via the MTT assay. The most potent platinum(IV) derivatives, **5** and **6**, which incorporated ACE, elicited the lowest GI_50_ values. Complexes **5** and **6** were found to be the most lipophilic complexes, and their high lipophilicity may have directly contributed to their exceptional potency. All the synthesised platinum(IV) complexes demonstrated exceptional in vitro activity compared with conventional chemotherapeutics, such as cisplatin, oxaliplatin and oxaliplatin. Despite the notable potency exhibited by **1**–**6**, they are not entirely selective towards cancer cells because they were also found to be toxic in the normal breast cancer cell line, MCF10A. Nevertheless, **1**–**6** may overcome drug resistance mechanisms associated with the cisplatin-resistant ADDP ovarian variant cell line. Complexes **1**–**6** also demonstrated both antioxidant and pro-oxidant effects, which were predominantly influenced by the intracellular release of the axial ligands, NPX and ACE, in the HT29 colon cells. The impact of the increased accumulation of ROS exhibited by the complexes at 24 h was observed in the mitochondrial activity, with the MtMP changes being significantly reduced. Furthermore, the COX-2 inhibitory activity of **1**–**6** was also assessed, with the results attesting that these complexes can reduce tumour-related inflammation, despite the lack of correlation between COX-2 inhibition and in vitro cytotoxicity and lipophilicity. Finally, cellular uptake studies for the most potent platinum(IV) complexes, **5** and **6**, are scheduled in the near future to further confirm whether enhanced cytotoxicity correlates with increased cellular uptake.

## 5. Patents

This work is part of Australian PCT application (**PCT/AU2023/050027**), Platinum(IV) complexes, 20 February 2023, Western Sydney University, Sydney, Australia.

## Figures and Tables

**Figure 1 cancers-15-02460-f001:**
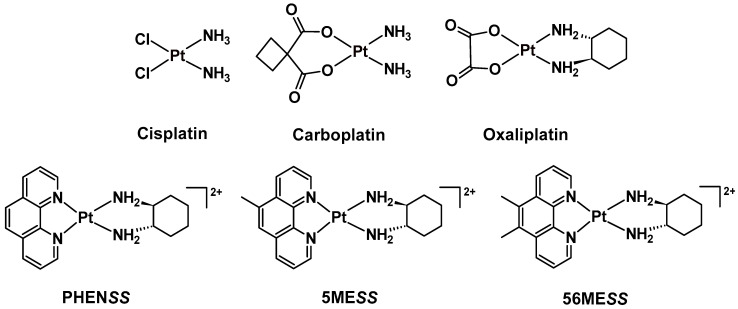
Structures of classical DNA-coordinating platinum(II) drugs, together with a non-traditional class of biologically active platinum(II) precursors of type **[Pt^II^(H_L_)(A_L_)]^2+^** (**PHEN*SS***, **5ME*SS*** and **56ME*SS***). Counter-ions were omitted for clarity.

**Figure 2 cancers-15-02460-f002:**
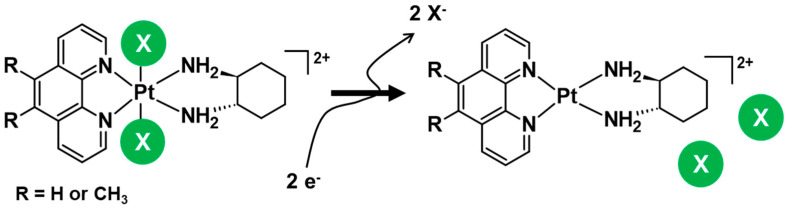
Exemplar platinum(IV) complex incorporating the platinum(II) scaffolds (**PHEN*SS***, **5ME*SS*** or **56ME*SS***) with axial ligands (**X**). The separation of the axial ligands from the platinum(II) scaffolds is initiated by the two-electron reduction process. Counter-ions were omitted for clarity.

**Figure 3 cancers-15-02460-f003:**
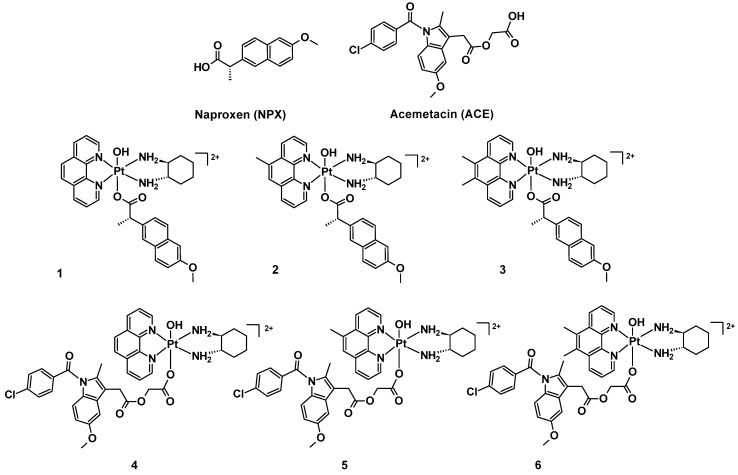
Chemical structures of NSAIDs, NPX and ACE, together with their platinum(IV) complexes, **1**–**6**. Counter-ions were omitted for clarity.

**Figure 4 cancers-15-02460-f004:**
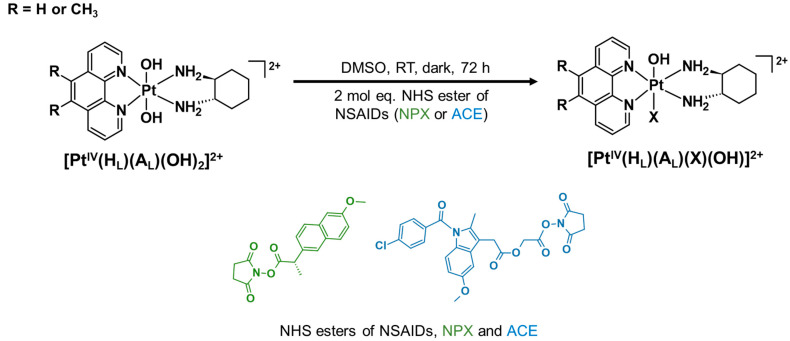
Synthesis route for **1**–**6**. Counter-ions were omitted for clarity.

**Figure 5 cancers-15-02460-f005:**
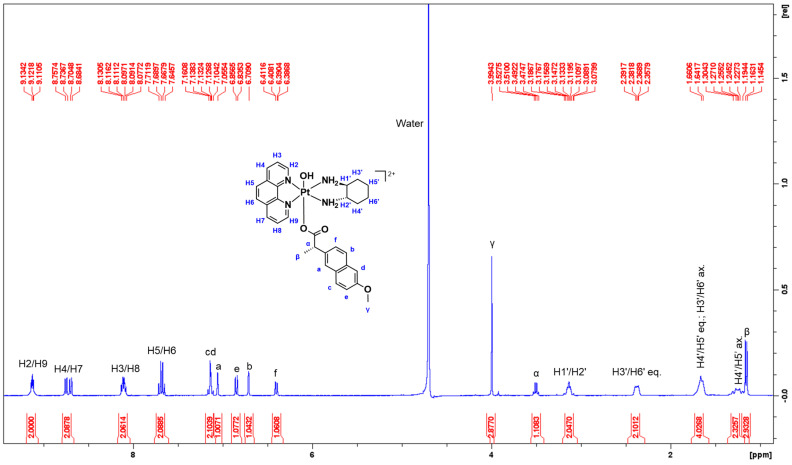
^1^H-NMR spectrum of **1** in D_2_O obtained at 298 K, with proton assignment. Inset: structure of **1** with proton labelling system.

**Figure 6 cancers-15-02460-f006:**
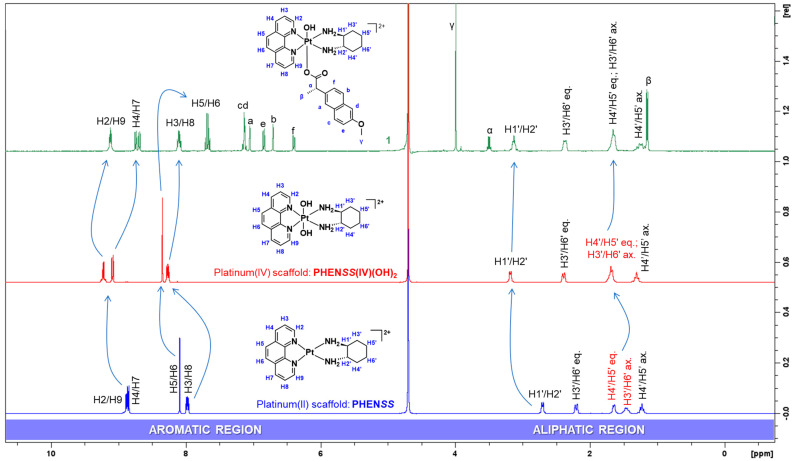
^1^H-NMR spectra of **PHEN*SS***, **PHEN*SS*(IV)(OH)_2_** and **1** in D_2_O obtained at 298 K, with arrows highlighting the change in multiplicity and movement of chemical resonances. Inset: structures of **PHEN*SS***, **PHEN*SS*(IV)(OH)_2_** and **1** with proton labelling systems.

**Figure 7 cancers-15-02460-f007:**
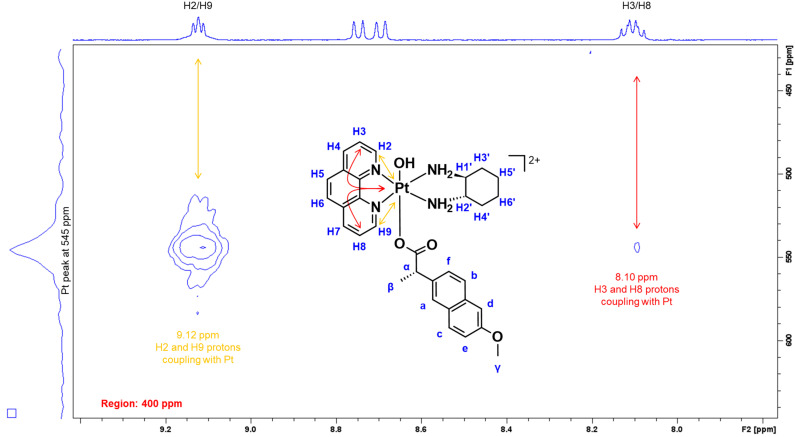
Expanded ^1^H-^195^Pt-HMQC spectrum of **1**, highlighting the correlation between H2, H9, H3 and H8 protons with the platinum core. Region: 400 ppm. Inset: structure of **1** with proton labelling system and arrows indicating correlation.

**Figure 8 cancers-15-02460-f008:**
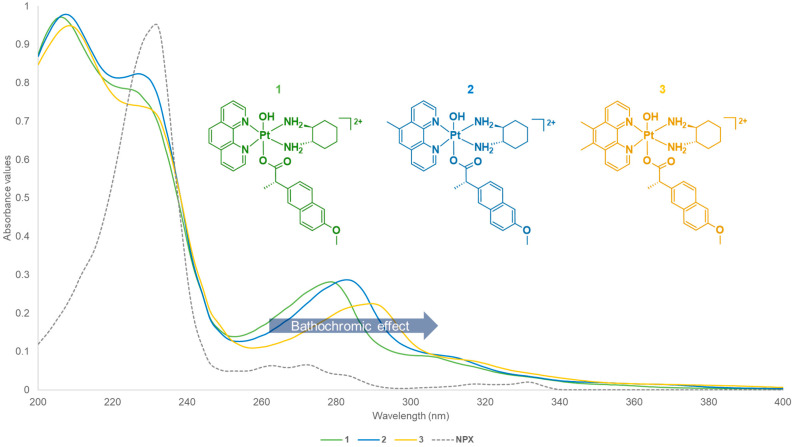
UV spectra of platinum(IV) complexes incorporating NPX (**1**–**3**) including the NPX ligand, obtained at 298 K, highlighting UV absorptions at different wavelengths. Inset: colour-coded structures of **1**–**3**.

**Figure 9 cancers-15-02460-f009:**
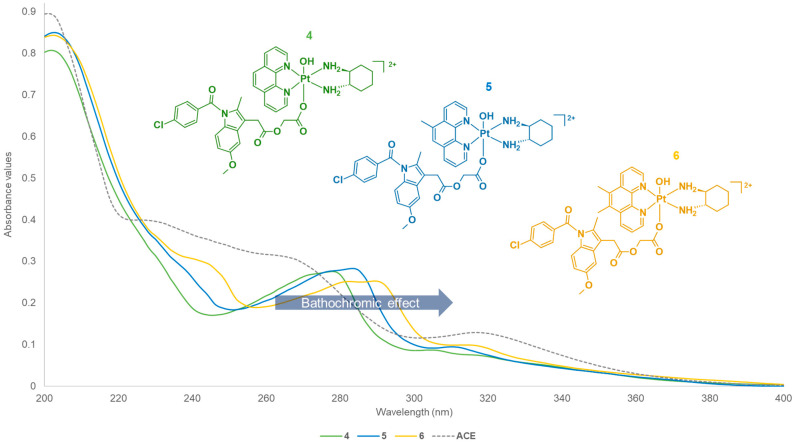
UV spectra of platinum(IV) complexes incorporating ACE (**4**–**6**) including the ACE ligand, obtained at 298 K, highlighting UV absorptions at different wavelengths. Inset: colour-coded structures of **4**–**6**.

**Figure 10 cancers-15-02460-f010:**
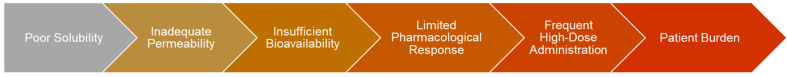
A flowchart showcasing the cumulative effects of administering a poorly soluble drug.

**Figure 11 cancers-15-02460-f011:**
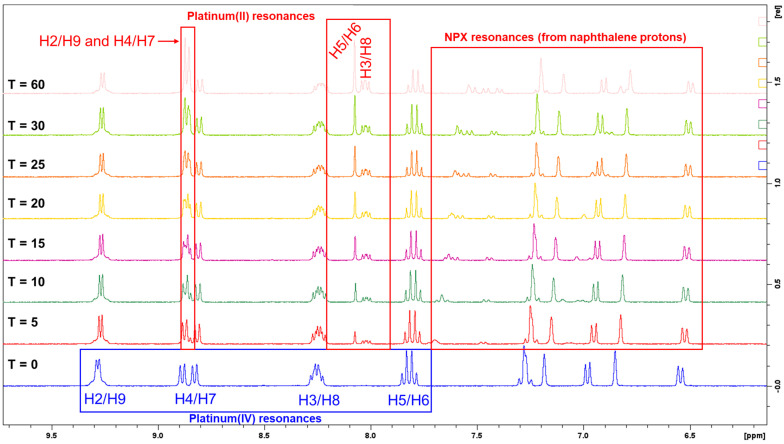
Expanded ^1^H-NMR reduction spectra of **1** with PBS and AsA in D_2_O at 37 °C, in different time intervals, showing the change in multiplicity from the phen protons and the naphthalene protons of the NPX ligand. **T** represents time in min.

**Figure 12 cancers-15-02460-f012:**
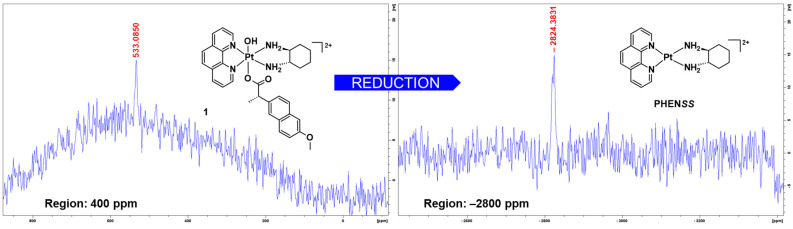
1D-^195^Pt-NMR reduction spectra of **1** with PBS and AsA in D_2_O, within the regions of 400 and −2800 ppm at 37 °C. The reduction of **1** to its platinum(II) precursor, **PHEN*SS***, is shown. Inset: structures of **1** and **PHEN*SS***.

**Figure 13 cancers-15-02460-f013:**
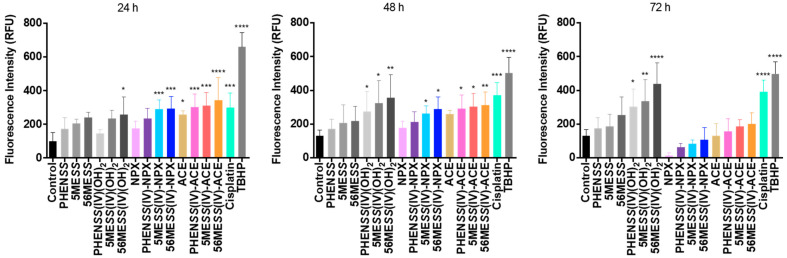
ROS production upon treatment with **1**–**6** (**PHEN*SS*(IV)-NPX** (**1**), **5ME*SS*(IV)-NPX** (**2**), **56ME*SS*(IV)-NPX** (**3**), **PHEN*SS*(IV)-ACE** (**4**), **5ME*SS*(IV)-ACE** (**5**), and **56ME*SS*(IV)-ACE** (**6**)), NPX, ACE, the platinum(II) precursors (**PHEN*SS***, **5ME*SS*** and **56ME*SS***) and platinum(IV) scaffolds (**PHEN*SS*(IV)(OH)_2_**, **5ME*SS*(IV)(OH)_2_** and **56ME*SS*(IV)(OH)_2_**), and cisplatin in HT29 colon cells at 24, 48 and 72 h. **** indicates *p* < 0.0001 compared with control. *** indicates *p* < 0.001 compared with control. ** indicates *p* < 0.01 compared with control. * indicates *p* < 0.05 compared with control. Data points denote mean ± SEM. *n* = 3 from three independent experiments where samples were run in triplicate. The data presented for platinum(II) precursors (**PHEN*SS***, **5ME*SS*** and **56ME*SS***) and platinum(IV) scaffolds (**PHEN*SS*(IV)(OH)_2_**, **5ME*SS*(IV)(OH)_2_** and **56ME*SS*(IV)(OH)_2_**) were obtained from previous studies [53,55].

**Figure 14 cancers-15-02460-f014:**
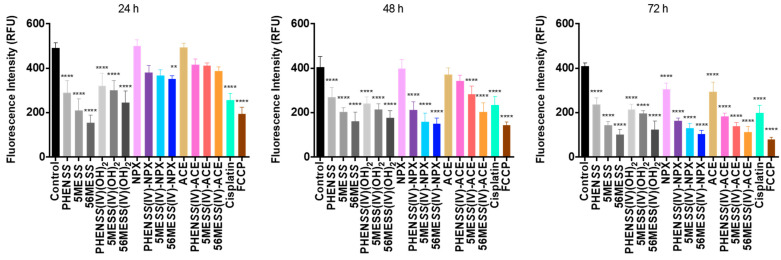
MtMP changes upon treatment with **1**–**6** (**PHEN*SS*(IV)-NPX** (**1**), **5ME*SS*(IV)-NPX** (**2**), **56ME*SS*(IV)-NPX** (**3**), **PHEN*SS*(IV)-ACE** (**4**), **5ME*SS*(IV)-ACE** (**5**), and **56ME*SS*(IV)-ACE** (**6**)), ACE, NPX, the platinum(II) precursors (**PHEN*SS***, **5ME*SS*** and **56ME*SS***) and platinum(IV) scaffolds (**PHEN*SS*(IV)(OH)_2_**, **5ME*SS*(IV)(OH)_2_** and **56ME*SS*(IV)(OH)_2_**), as well as cisplatin in HT29 colon cells at 24, 48 and 72 h. **** indicates *p* < 0.0001 compared with control. ** indicates *p* < 0.01 compared with control. Data points denote mean ± SEM. *n* = 3 from three independent experiments where samples were run in triplicate.

**Figure 15 cancers-15-02460-f015:**
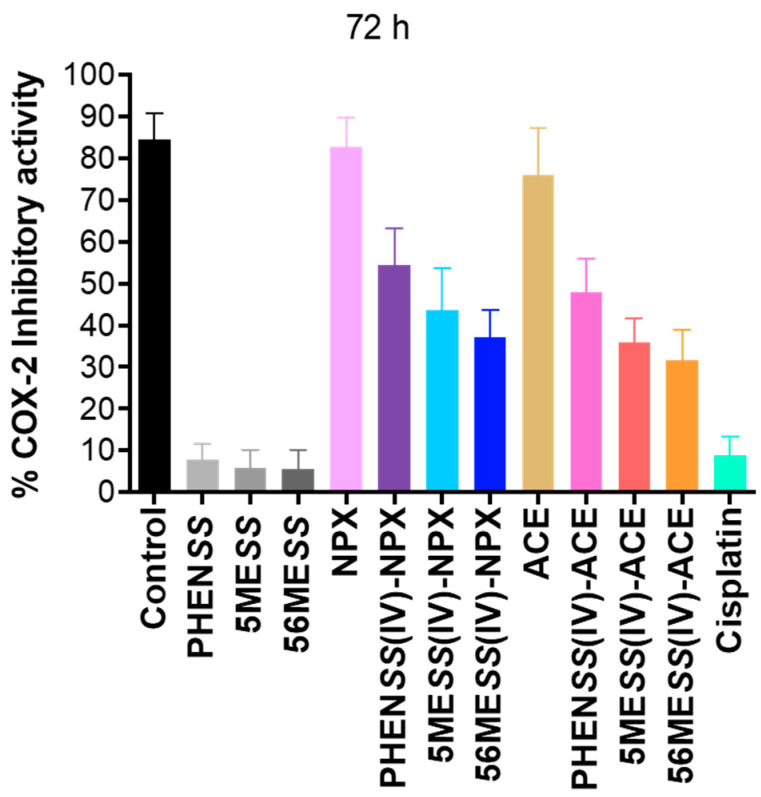
COX-2 inhibition of **1**–**6** (**PHEN*SS*(IV)-NPX** (**1**), **5ME*SS*(IV)-NPX** (**2**), **56ME*SS*(IV)-NPX** (**3**), **PHEN*SS*(IV)-ACE** (**4**), **5ME*SS*(IV)-ACE** (**5**), and **56ME*SS*(IV)-ACE** (**6**)), ACE, NPX, the platinum(II) precursors (**PHEN*SS***, **5ME*SS*** and **56ME*SS***), and cisplatin. Data points denote mean ± SEM. *n* = 3 from three independent experiments where samples were run in triplicate.

**Table 1 cancers-15-02460-t001:** A summary of experimental yields (%), HPLC peak areas (%), T_R_ (min) and mass-to-charge ratios (*m*/*z*) of **1**–**6**.

Platinum(IV) Complexes	Experimental Yields (%)	T_R_ (min)	HPLC Peak Areas (%)	Mass-to-Charge Ratios (*m*/*z*)	
				Calc.	Exp.
**1**	60.0	7.78	97	734.23	734.23
**2**	69.0	7.98	98	748.25	748.25
**3**	65.0	8.19	99	762.26	762.26
**4**	62.0	9.62	97	919.22	920.22
**5**	68.0	9.83	96	933.23	934.22
**6**	58.0	10.2	97	947.25	948.25

Calc., calculated; Exp., experimental.

**Table 2 cancers-15-02460-t002:** Summary of the ^1^H-NMR and ^1^H-^195^Pt-HMQC data of platinum(IV)-NPX derivatives (**1**–**3**), including chemical shifts (δ ppm), multiplicity, integration and *J*-coupling constants (Hz).

Proton Labels	1	2	3
H2	9.12 (t, 2H, *J* = 5.0 Hz) (signals overlapping)	9.12 (t, 1H, *J* = 5.6 Hz)	9.06 (d, 2H, *J* = 5.5 Hz) (signals overlapping)
H9		9.04 (d, 1H, *J* = 5.6 Hz)	
H4	8.74 (d, 1H, *J =* 8.3 Hz)	8.84 (q, 1H, *J =* 8.5 Hz)	8.84 (t, 2H, *J* = 9.0 Hz) (signals overlapping)
H7	8.70 (d, 1H, *J* = 8.3 Hz)	8.60 (q, 1H, *J =* 8.3 Hz)	
H5 and H6	7.67 (q, 2H, *J =* 8.9 Hz)	-	-
H6	-	7.40 (d, 1H, *J* = 19 Hz)	-
H3	8.10 (m, 2H) (signals overlapping)	8.10 (m, 2H) (signals overlapping)	8.09 (m, 2H) (signals overlapping)
H8			
a	7.05 (s, 1H)	6.67 (d, 1H, *J =* 7.8 Hz)	6.94 (s, 1H)
b	6.71 (s, 1H)	6.40 (d, 1H, *J =* 8.4 Hz)	6.62 (s, 1H)
c	7.12 (m, 2H) (signals overlapping)	7.10 (d, 2H, *J =* 3.2 Hz) (signals overlapping)	7.01 (s, 2H) (signals overlapping)
d			
e	6.84 (d, 1H, *J* = 8.5 Hz)	7.03 (d, 1H, *J* = 4.5 Hz)	6.77 (d, 1H, *J* = 8.5 Hz)
f	6.40 (d, 1H, *J*_1_ = 1.4, *J*_2_ = 8.5 Hz)	6.84 (dd, 1H, *J*_1_ *=* 2.4, *J*_2_ = 8.5 Hz)	6.38 (d, 1H, *J =* 8.4 Hz)
γ	3.99 (s, 3H)	2.56 (s, 3H)	3.92 (s, 3H)
α	3.50 (m, 1H, *J* = 7.0 Hz)	3.50 (m, 1H)	3.48 (q, 1H, *J =* 7.0 Hz)
H1′ and H2′	3.12 (m, 2H)	3.14 (d, 2H)	3.16 (t, 2H)
CH_3_ (5 position)	-	3.97 (s, 3H)	3.92 (s, 3H)
2 × CH_3_ (5 and 6 positions)	-	-	2.42 (s, 8H) (signals overlapping)
H3′ and H6′ eq.	2.37 (m, 2H)	2.38 (d, 2H)	
H4′ and H5′ eq.; H3′ and H6′ ax.	1.65 (m, 4H)	1.66 (m, 4H)	1.67 (m, 4H)
H4′ and H5′ ax.	1.25 (m, 2H)	1.27 (m, 2H)	1.28 (m, 2H)
β	1.15 (d, 3H, *J* = 7.1 Hz)	1.15 (q, 3H, *J* = 4.6 Hz)	1.15 (d, 3H, *J* = 7.0 Hz)
^1^H/^195^Pt	9.12, 8.10/545	9.12, 9.04/544	9.06, 8.09/534

**Table 3 cancers-15-02460-t003:** Summary of the ^1^H-NMR and ^1^H-^195^Pt-HMQC data of platinum(IV)-ACE derivatives (**4**–**6**), including chemical shifts (δ ppm), multiplicity, integration and *J*-coupling constants (Hz).

Proton Labels	4	5	6
H2	9.16 (d, 1H, *J =* 5.4 Hz)	9.18 (dd, 1H, *J =* 5.5 Hz)	9.12 (d, 1H, *J =* 5.4 Hz)
H9	9.14 (d, 1H, *J =* 5.5 Hz)	9.06 (dd, 1H, *J =* 5.4 Hz)	9.06 (d, 1H, *J =* 5.4 Hz)
H4	8.68 (q, 2H, *J =* 5.6 Hz)	8.79 (q, 1H, *J =* 5.0 Hz)	8.76 (q, 2H, *J =* 4.9 Hz)
H7		8.41 (q, 1H, *J =* 8.5 Hz)	
H5 and H6	7.91 (s, 2H)	-	-
H6	-	7.58 (d, 1H, *J =* 6.4 Hz)	-
H3	8.03 (q, 2H, *J =* 5.8 Hz) (signals overlapping)	8.08 (t, 1H, *J* = 6.8 Hz)	7.98 (m, 1H)
H8		7.92 (m, 1H)	
a and b	7.21 (s, 4H) (signals overlapping)	7.05 (q, 4H, *J =* 8.0 Hz) (signals overlapping)	6.95 (s, 4H) (signals overlapping)
c and d			
a1	6.53 (s, 1H)	6.52 (s, 1H)	6.53 (s, 2H) (signals overlapping)
b1	6.69 (d, 1H, *J =* 8.4 Hz)	6.58 (d, 1H, *J* = 8.7 Hz)	
c1	6.42 (d, 1H, *J =* 8.4 Hz)	6.27 (d, 1H, *J* = 8.6 Hz)	6.18 (d, 1H, *J* = 7.6 Hz)
f	3.55 (s, 3H)	3.48 (s, 3H)	3.44 (s, 3H)
α2	4.27 (s, 2H)	4.28 (m, 2H)	4.29 (s, 2H)
α1	3.37 (s, 2H)	3.34 (s, 2H)	3.33 (s, 2H)
H1′ and H2′	3.15 (s, 2H)	3.15 (d, 2H)	3.14 (m, 2H)
CH_3_ (5 position)	-	2.50 (s, 3H)	-
2 × CH_3_ (5 and 6 positions)	-	-	2.38 (s, 8H) (signals overlapping)
H3′ and H6′ eq.	2.33 (s, 2H)	2.34 (d, 2H)	
e	1.76 (s, 3H)	1.65 (m, 7H) (signals overlapping)	1.62 (m, 7H) (signals overlapping)
H4′ and H5′ eq.; H3′ and H6′ ax.	1.65 (m, 4H)		
H4′ and H5′ ax.	1.21 (m, 2H)	1.23 (m, 2H)	1.23 (m, 2H)
β	-	-	-
^1^H/^195^Pt	9.16, 9.14, 8.03/547	9.18, 9.06, 8.08, 7.92/545	9.12, 9.09, 7.98/531

**Table 4 cancers-15-02460-t004:** Characteristic peaks in the UV and CD spectra of **1**–**6**.

Platinum (IV) Complexes	UV λ_max_ nm (ε/M.cm^−1^ ± SD × 10^4^)	CD λ_max_ nm (Δε/M.cm^−1^ × 10^1^)
**1**	206 (8.92 ± 5.54), 227 (7.08 ± 3.59), 279 (2.56 ± 0.25), 306 (0.77 ± 0.88)	202 (−1263), 224 (+178), 239 (+329), 281 (−137)
**2**	207 (9.95 ± 6.78), 228 (8.39 ± 4.98), 283 (2.92 ± 0.12), 312 (0.86 ± 0.85)	201 (−1061), 223 (+120), 232 (+48.1), 243 (+309), 287 (−109)
**3**	208 (8.73 ± 5.35), 231 (6.74 ± 3.23), 289 (2.08 ± 0.42), 318 (0.67 ± 0.91)	204 (−1007), 223 (+9.32), 230 (−9.99), 244 (+316), 293 (−93.2)
**4**	203 (4.43 ± 0.42), 278 (1.15 ± 2.70), 305 (0.49 ± 3.40)	206 (−447), 249 (+8.15)
**5**	204 (18.6 ± 5.44), 284 (6.16 ± 2.63), 311 (2.12 ± 1.68)	205 (−373), 209 (−419), 234 (−144)
**6**	204 (16.7 ± 4.75), 244 (5.90 ± 2.34), 290 (5 ± 1.40), 316 (1.94 ± 0.47)	214 (−175), 241 (−29.2), 285 (+11.9)

**Table 5 cancers-15-02460-t005:** Characteristic peaks in the IR spectra of **1**–**6**.

Platinum(IV) Complexes	Type of Bond and Frequency (cm^−1^)					
	O-H Stretch (Alcohol)	C-H Stretch (Aromatic)	C-H Stretch (Alkyl)	C-C Stretch (Aromatic)	C-O Stretch (Alkyl Aryl Ether)	C-Cl Stretch (Halogen)
**1**	3380	3062	2936	1603	1220	-
**2**	3471	3076	2940	1605	1220	-
**3**	3466	3076	2937	1603	1218	-
**4**	3410	3063	2942	1609	1262	845
**5**	3385	3066	2943	1608	1265	846
**6**	3418	3071	2941	1609	1263	846

**Table 6 cancers-15-02460-t006:** Solubility of **1**–**6** in d.i.H_2_O at room temperature, with values expressed in mg/mL and mol/L.

Complexes	Solubility	
	mol/L	mg/mL
**1**	3.7 × 10^−2^	32
**2**	3.2 × 10^−2^	28
**3**	4.3 × 10^−2^	38
**4**	1.5 × 10^−2^	26
**5**	2.9 × 10^−2^	31
**6**	3.3 × 10^−2^	36
Cisplatin	-	2.5 *
Oxaliplatin	-	7.9 *
Carboplatin	-	11.7 *

* The solubility of cisplatin, oxaliplatin and carboplatin were obtained from the literature [76].

**Table 7 cancers-15-02460-t007:** A summary of log k_w_ values of **1**–**6**.

Complexes	Heterocyclic Ligands (or H_L_)	NSAID Axial Ligands	log k_w_ Values
**1**	Phen	NPX	1.44
**2**	5-Mephen	NPX	1.46
**3**	5,6-Me_2_phen	NPX	1.48
**4**	Phen	ACE	1.93
**5**	5-Mephen	ACE	2.16
**6**	5,6-Me_2_phen	ACE	2.29

**Table 8 cancers-15-02460-t008:** A summary of estimated time points in min for **1**–**6** at which 50 and 100% reductions proceed, expressed as T_50%_ and T_100%_.

Platinum(IV) Complexes	Heterocyclic Ligands (or H_L_)	NSAID Axial Ligands	Reduction Times (min)	
			T_50%_	T_100%_
**1**	Phen	NPX	20–25	post 60
**2**	5-Mephen	NPX	10–15	post 60
**3**	5,6-Me_2_phen	NPX	10–15	post 60
**4**	Phen	ACE	0–5	10–15
**5**	5-Mephen	ACE	0–5	10–15
**6**	5,6-Me_2_phen	ACE	0–5	5

**Table 9 cancers-15-02460-t009:** A summary of the GI_50_ values (nM) of **1**–**6**, NPX and ACE in multiple cell lines. The data presented for cisplatin, oxaliplatin, carboplatin and the precursor platinum(II) and platinum(IV) scaffolds were obtained from previous studies [53,55]. *nd*, not determined; SCI, GI_50_ in MCF10A/GI_50_ in Du145.

*Platinum(IV)* *Prodrugs*	HT29	U87	MCF7	A2780	H460	A431	Du145	BE2C	SJG2	MIA	MCF10A	ADDP	Mean GI_50_ Values	Selectivity Cytotoxicity Index (SCI)
**GI_50_ Values (nM)**														
**1**	400 ± 23	2570 ± 440	1120 ± 342	420 ± 50	860 ± 124	1020 ± 390	230 ± 17	1470 ± 67	1570 ± 67	660 ± 100	450 ± 24	400 ± 19	931 ± 139	1.96
**2**	210 ± 19	1330 ± 190	720 ± 240	290 ± 56	410 ± 37	530 ± 180	130 ± 27	680 ± 17	620 ± 43	370 ± 54	310 ± 12	280 ± 27	490 ± 75	2.38
**3**	28 ± 6	180 ± 23	60 ± 22	39 ± 9	56 ± 5	75 ± 33	10 ± 2	220 ± 12	230 ± 18	47 ± 7	36 ± 4	25 ± 2	84 ± 12	3.60
**4**	290 ± 30	1720 ± 140	840 ± 390	350 ± 47	610 ± 80	930 ± 490	150 ± 40	1300 ± 150	1450 ± 104	490 ± 36	360 ± 29	330 ± 27	735 ± 58	2.40
**5**	13 ± 8	140 ± 8	39 ± 18	30 ± 12	43 ± 8	60 ± 45	1.3 ± 0.4	250 ± 33	200 ± 21	29 ± 14	20 ± 5	9.3 ± 2	70 ± 15	15.4
**6**	4.7 ± 4	28 ± 3	12 ± 6	10 ± 4	16 ± 2	16 ± 12	0.22 ± 0.1	160 ± 19	130 ± 25	7.8 ± 4	3.4 ± 1	1.3 ± 0.5	32 ± 7	15.5
** *Unconventional Platinum(II) Complexes* **														
**PHEN*SS***	160 ± 45	980 ± 270	1500 ± 500	230 ± 30	360 ± 35	480 ± 170	100 ± 38	380 ± 46	330 ± 66	200 ± 57	300 ± 58	190 ± 47	434 ± 110	3.00
**5ME*SS***	33 ± 4	320 ± 26	200 ± 12	61 ± 10	41 ± 5	120 ± 25	22 ± 3	270 ± 38	220 ± 10	48 ± 2	30 ± 2	34 ± 2	117 ± 12	1.36
**56ME*SS***	10 ± 1.6	35 ± 6.4	93 ± 44	76 ± 57	21 ± 2	29 ± 1	4.6 ± 0.4	59 ± 4	66 ± 22	13 ± 2	16 ± 1	13 ± 2	36 ± 10	3.48
** *Unconventional Platinum(IV) Scaffolds* **														
**PHEN*SS*(IV)(OH)_2_**	710 ± 300	4900 ± 610	16,000 ± 4500	800 ± 84	1700 ± 200	4300 ± 530	310 ± 92	3000 ± 530	1700 ± 350	3400 ± 2200	1700 ± 200	1300 ± 350	3318 ± 880	5.48
**5ME*SS*(IV)(OH)_2_**	60 ± 6	900 ± 58	1200 ± 390	240 ± 9	60 ± 5	360 ± 58	41 ± 5	1400 ± 300	640 ± 70	160 ± 29	130 ± 19	130 ± 22	443 ± 81	3.17
**56ME*SS*(IV)(OH)_2_**	36 ± 7	190 ± 23	480 ± 140	59 ± 7	190 ± 150	120 ± 22	15 ± 2.6	240 ± 22	210 ± 45	43 ± 2.5	61 ± 7	170 ± 120	151 ± 50	4.07
** *Traditional Chemotherapy Agents* **														
Cisplatin	11,300 ± 1900	3800 ± 1100	6500 ± 800	1000 ± 100	900 ± 200	2400 ± 300	1200 ± 100	1900 ± 200	400 ± 100	7500 ± 1300	5200 ± 520	28,000 ± 1600	5842 ± 610	4.33
Oxaliplatin	900 ± 200	1800 ± 200	500 ± 100	160 ± 100	1600 ± 100	4100 ± 500	2900 ± 400	900 ± 200	3000 ± 1200	900 ± 200	*nd*	800 ± 100	1463 ± 320	*nd*
Carboplatin	>50,000	>50,000	>50,000	9200 ± 2900	14,000 ± 1000	24,000 ± 2200	15,000 ± 1200	19,000 ± 1200	5700 ± 200	>50,000	>50,000	>50,000	32,242 ± 1450	3.33
** *NSAIDs (Axial Ligands)* **														
NPX	>50,000	>50,000	>50,000	>50,000	>50,000	>50,000	>50,000	>50,000	>50,000	>50,000	>50,000	>50,000	>50,000	1.00
ACE	>50,000	>50,000	>50,000	>50,000	>50,000	>50,000	>50,000	>50,000	>50,000	>50,000	>50,000	>50,000	>50,000	1.00

**Table 10 cancers-15-02460-t010:** Production of ROS upon treatment with **1**–**6**, NPX, ACE, and cisplatin in HT29 colon cells at 24, 48 and 72 h. The data presented for platinum(II) precursors (**PHEN*SS***, **5ME*SS*** and **56ME*SS***) and platinum(IV) scaffolds (**PHEN*SS*(IV)(OH)_2_**, **5ME*SS*(IV)(OH)_2_** and **56ME*SS*(IV)(OH)_2_**) were obtained from previous studies [53,55].

Compounds	ROS Production in Different Time Intervals (RFU)		
	24 h	48 h	72 h
Control	100 ± 17	130 ± 11	131 ± 12
Cisplatin	299 ± 25	370 ± 22	392 ± 21
TBHP	658 ± 28	504 ± 15	497 ± 10
**PHEN*SS***	174 ± 2	172 ± 9	176 ± 7
**5ME*SS***	204 ± 4	205 ± 3	188 ± 3
**56ME*SS***	240 ± 5	218 ± 3	255 ± 4
**PHEN*SS*(IV)(OH)_2_**	144 ± 5	273 ± 4	303 ± 1
**5ME*SS*(IV)(OH)_2_**	234 ± 1	323 ± 9	335 ± 2
**56ME*SS*(IV)(OH)_2_**	259 ± 3	356 ± 11	438 ± 7
NPX	187 ± 12	178 ± 13	15 ± 5
**1**	235 ± 23	215 ± 19	63 ± 7
**2**	290 ± 17	261 ± 15	84 ± 9
**3**	297 ± 21	287 ± 11	104 ± 15
ACE	279 ± 7	260 ± 9	132 ± 13
**4**	303 ± 8	298 ± 17	168 ± 15
**5**	313 ± 11	305 ± 10	192 ± 7
**6**	344 ± 17	312 ± 13	202 ± 10

**Table 11 cancers-15-02460-t011:** MtMP changes upon treatment with **1**–**6**, NPX, ACE, the platinum(II) precursors (**PHEN*SS***, **5ME*SS*** and **56ME*SS***) and platinum(IV) scaffolds (**PHEN*SS*(IV)(OH)_2_**, **5ME*SS*(IV)(OH)_2_** and **56ME*SS*(IV)(OH)_2_**), and cisplatin in HT29 colon cells at 24, 48 and 72 h.

Compounds	MtMP Changes in Different Time Intervals (RFU)		
	24 h	48 h	72 h
Control	495 ± 7	412 ± 15	411 ± 5
Cisplatin	256 ± 9	236 ± 13	199 ± 11
FCCP	194 ± 10	143 ± 5	80 ± 5
**PHEN*SS***	289 ± 17	271 ± 15	237 ± 10
**5ME*SS***	210 ± 15	204 ± 6	144 ± 5
**56ME*SS***	156 ± 10	162 ± 13	103 ± 7
**PHEN*SS*(IV)(OH)_2_**	321 ± 17	241 ± 10	216 ± 8
**5ME*SS*(IV)(OH)_2_**	301 ± 14	214 ± 9	198 ± 4
**56ME*SS*(IV)(OH)_2_**	245 ± 16	177 ± 11	125 ± 12
NPX	499 ± 10	397 ± 13	304 ± 9
**1**	382 ± 11	211 ± 12	163 ± 5
**2**	368 ± 9	160 ± 13	130 ± 7
**3**	351 ± 5	152 ± 8	104 ± 6
ACE	494 ± 6	371 ± 10	295 ± 13
**4**	416 ± 8	343 ± 8	184 ± 5
**5**	412 ± 4	284 ± 12	140 ± 5
**6**	387 ± 7	284 ± 11	112 ± 8

**Table 12 cancers-15-02460-t012:** The % of COX-2 inhibition of **1**–**6**, NPX, ACE, the platinum(II) precursors (**PHEN*SS***, **5ME*SS*** and **56ME*SS***), and cisplatin at 72 h.

Compounds	Percentage COX-2 Inhibitory Activity (%)
	72 h
Control	84 ± 3
Cisplatin	9 ± 2
**PHEN*SS***	8 ± 4
**5ME*SS***	6 ± 3
**56ME*SS***	5 ± 4
NPX	83 ± 5
**1**	55 ± 3
**2**	44 ± 6
**3**	37 ± 4
ACE	76 ± 8
**4**	48 ± 3
**5**	36 ± 2
**6**	32 ± 3

## Data Availability

All data relevant to the publication are included.

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
