# Peer review of "Versatile Platinum(IV) Prodrugs of Naproxen and Acemetacin as Chemo-Anti-Inflammatory Agents"

_cancers, 2023, doi:10.3390/cancers15092460_

Round 1

Reviewer 1 Report

The authors describe the synthesis, characterization and biological properties of six Pt(IV) complexes containing NSAIDs (NPX and ACE) as ligands. The work is done excellently and without any problems. I may recommend that manuscript for publication as it is.

Only three minor points:

line 337/374: "aromatic protons" change by "protons" (protons are not aromatic, that is lab jargon)

line 374: In my opinion, "195Pt peaks" is lab jargon, too

A recently published paper on that topic could be cited in the introduction: Int. J. Mol. Sci. 2023, 24, 5718. https://doi.org/10.3390/ijms24065718

Reviewer 2 Report

This manuscript describes the synthesis and characterization of six mono-substituted platinum(IV) complexes incorporating the NSAIDs, NPX and ACE compounds along with the their antitumour activity evaluation. Authors conducted a big number of experiments/assays concerning the biological evaluation of 6 compounds and it is a significant research effort. However, there are many drawbacks (mainly concerning the chemical structure confirmation/characterization of the complexes) that authors must addressed in order the manuscript to be suitable for publication.

The detailed NMR spectral analysis of the synthesized compounds accompanied with MS and HPLC may well verify the composition and homogeneity of the studied complexes, however, in terms of confirmation of the detailed chemical structure of the obtained inorganic platinum(IV) complexes, X-ray crystallography (along with IR analysis) is required. The lack of this data is the main drawback of this study. Authors must provide a more detailed chemical characterization of the complexes and also report the fluorescence nature of compounds after complexation with platinum(IV), bearing in mind that free ligands are fluorescent molecules. Authors conclude that the most cytotoxic platinum(IV) derivatives in the study were 5 and 6, a fact that may be correlated with the increased lipophilicity. However, there are many other parameters that have an impact on the biological activity of a compound like the charge, the chemical properties/nature of the incorporated (ligands 5-Mephen and 5,6-Me2phen heterocyclic ligands, respectively, and ACE as the axial ligand), thus authors must include an extensive analysis concerning the aforementioned data. Furthermore, a comparison of the biological activities of the studied complexes with those of similar studies of the literature must also be added in the manuscript. Even though compound 6 proved to be most cytotoxic in the studied cancer cells is also significantly toxic in the MCF-10A non-tumorigenic epithelial cell line, consequently there is no selectivity towards cancer and no-cancer cells and authors must also add a comment concerning this observation. Cell uptake analysis (if complexes are fluorescent) would be very helpful to conclude whether the increased toxicity correlates with increased uptake of complexes 5 and 6 (bearing in mind that are lipophilic and positively charged). In terms of uniformity, the name of cells lines employed must either reported with a dash symbol or not (for example, MCF-7, HT-29, etc.)

Reviewer 3 Report

This manuscript describes Platinum(IV) Prodrugs of Naproxen and Acemetacin. The results are interesting and very exciting. All compounds showed activity superior to the established platinum drugs. All compounds are fully characterised for the proof of the structure, but also in the terms of pharmacokinetics parameters. I think this manuscript can be accepted in the current form.

Round 2

Reviewer 2 Report

Authors addressed the majority of the mentioned points and the manuscript can be accepted for publication.